# Optimal Anytime Algorithms for Online Convex Optimization with Adversarial Constraints

**Dhruv Sarkar** [1]  **Abhishek Sinha** [2]

## Abstract

We propose an anytime online algorithm for learning a sequence of convex cost functions while approximately satisfying a sequence of convex constraints, without prior knowledge of the time horizon. Both the cost and constraint functions may be chosen adversarially over time. While this problem has recently been resolved in the setting where the time horizon is known, extending these guarantees to the anytime setting, without resorting to inefficient doubling tricks, has remained technically challenging. Our main contribution is the introduction of a time-varying yet horizon-oblivious Lyapunov function to track constraint violations. The use of such a time-varying Lyapunov function introduces new technical difficulties, as a key monotonicity property underlying prior analyses no longer holds. By developing a novel analytical technique, we show that our algorithm achieves $O(\sqrt{t})$ Regret and $\tilde{O}(\sqrt{t})$ cumulative constraint violation (CCV) for all $t \geq 1$. We further extend our framework to the dynamic regret setting, obtaining bounds that adapt to the unknown path length of the comparator sequence. Finally, we present an adaptive algorithm for the optimistic setting, whose performance scales gracefully with the cumulative prediction error. We validate the practical effectiveness of our approach through numerical experiments on the online shortest path problem.

## 1. Introduction

We consider a natural generalization of the standard Online Convex Optimization (OCO) problem, termed *Constrained Online Convex Optimization (COCO)*, in which, in addition to a convex cost function, a convex constraint function is revealed to the learner after it selects an action at each round. The objective is to design an online algorithm that achieves small Regret with respect to the cost functions while simultaneously ensuring small cumulative constraint violation (CCV) with respect to the constraint functions for all rounds $t \geq 1$. The COCO framework arises in a variety of practical applications, including safety-aware contextual bandits (Sun et al., 2017), autonomous driving (Gao et al., 2024), budget-constrained bandits (Immorlica et al., 2022), and learning safety constraints in large language models (Chen et al., 2025).

COCO has been extensively studied over the past decade (Guo et al., 2022; Yi et al., 2023; Neely & Yu, 2017). Recently, Sinha & Vaze (2024) proposed a simple algorithm that achieves $O(\sqrt{T})$ Regret and $\tilde{O}(\sqrt{T})$ cumulative constraint violation for a given horizon of length $T$, and showed that these rates are information-theoretically optimal when the horizon length $T$ is known. Their approach constructs a surrogate cost function by linearly combining the cost and constraint functions at each round, with coefficients determined by a Lyapunov function, and then applies a standard OCO algorithm to this surrogate loss. The resulting analysis is notably elegant, especially in contrast to earlier primal–dual approaches, which typically yield weaker guarantees (Guo et al., 2022; Yi et al., 2021; 2023; Sun et al., 2017).

However, a major limitation of Sinha & Vaze (2024)'s result is that the proposed algorithm is *not* anytime, as it requires prior knowledge of the horizon length $T$. In particular, although the algorithm achieves $O(\sqrt{t})$ Regret for all $t \geq 1$, the corresponding CCV guarantee scales as $\tilde{O}(\sqrt{T})$ for all $t$, regardless of how small $t$ is (Sinha & Vaze, 2024, Theorem 1). This leads to the following natural and important question: *Is it possible to design a COCO algorithm that simultaneously achieves $O(\sqrt{t})$ Regret and $\tilde{O}(\sqrt{t})$ CCV for all $t \geq 1$?*

In this paper, we answer this question in the affirmative by making use of a sequence of time-varying Lyapunov functions $\{\Phi_t(\cdot)\}_{t \geq 1}$, which replace the fixed, horizon-dependent Lyapunov function used in Sinha & Vaze (2024). Their analysis critically relies on the fact that the sequence $\{\Phi(Q(t))\}_{t \geq 1}$ is monotonically non-decreasing, where

---

[1]Indian Institute of Technology, Kharagpur [2]Tata Institute of Fundamental Research, Mumbai, India. Correspondence to: Dhruv Sarkar <dhruv.sarkar223@gmail.com>, Abhishek Sinha <abhishek.sinha.tifr@gmail.com>.

*Proceedings of the 43$^{rd}$ International Conference on Machine Learning*, Seoul, South Korea. PMLR 306, 2026. Copyright 2026 by the author(s).

$Q(t)$ denotes the cumulative constraint violation up to round $t$. However, with the new time-varying Lyapunov functions, the functions $\Phi_t(x)$ are themselves monotonically decreasing in $t$ for every fixed $x$, which invalidates the original argument.

Our key technical contribution is to show that, despite this apparent difficulty, the sequence $\{\Phi_t(Q(t))\}_{t \geq 1}$ can still be made monotonically non-decreasing, and hence one can establish the performance bounds using a more sophisticated analysis. This is achieved by defining $Q(t)$ as a carefully chosen *upper bound* on the true cumulative constraint violation and by exploiting the exponential form of the Lyapunov functions (see Eqn. (7)). Importantly, our approach completely avoids the unwieldy doubling trick, which is known to suffer from poor practical performance due to discarding a substantial amount of information (Zhang et al., 2024; Kwon & Mertikopoulos, 2014; Luo & Schapire, 2014).

In summary, we make the following contributions in this paper:

- We propose a simple *anytime* algorithm for COCO that achieves $O(\sqrt{t})$ Regret and $\widetilde{O}(\sqrt{t})$ cumulative constraint violation (CCV) for all rounds $t \geq 1$. Our algorithm does not require restarts and completely avoids the doubling trick (see Section A). We also derive adaptive bounds under standard first-order feedback when the Lipschitz constant is unknown; the remaining dependence on the diameter $D$ is discussed in Remark 4 and Section 8.

- Our results are established by introducing a sequence of *time-varying Lyapunov functions* that are oblivious to the horizon length. In contrast to prior work based on time-invariant Lyapunov functions, the time-varying setting introduces fundamental technical challenges due to the loss of a key monotonicity property. We address this issue by working with a carefully chosen upper bound on the cumulative constraint violation and by appropriately tuning the Lyapunov parameters. This technique may be of independent interest.

- We further show that our framework extends naturally to the *dynamic regret* and *optimistic* settings. In particular, we obtain algorithms whose performance adapts to unknown problem-dependent quantities such as the path length ($\mathcal{P}_T$) and the cumulative prediction error ($\mathcal{E}_T$), without resorting to the doubling trick.

- Finally, we demonstrate the practical effectiveness of our approach through a series of numerical experiments.

## 2. Related Work

The study of online learning with long-term constraints was initiated by Mannor et al. (2009). In the context of two-player infinite-horizon stochastic games, they established a fundamental impossibility result: it is impossible to simultaneously achieve sublinear bounds for both Regret and cumulative constraint violation (CCV) against the best fixed offline action that satisfies the long-term constraint over the entire horizon. This negative result motivated subsequent works to consider a weaker benchmark, in which the benchmark satisfies the constraints at *every* round (Mahdavi et al., 2012; Neely & Yu, 2017; Guo et al., 2022). For time-invariant constraints, Mahdavi et al. (2012) used Online Gradient Descent and mirror-prox policies to obtain sublinear guarantees. Later, Castiglioni et al. (2022) proposed a unified meta-algorithm with $\mathcal{O}(T^{3/4})$ approximate regret and CCV bounds under additional assumptions, including Slater's condition. For adversarial constraints, Guo et al. (2022) achieved $O(\sqrt{T})$ Regret and $O(T^{3/4})$ CCV without Slater's condition. The closest work to ours is Sinha & Vaze (2024), who obtain optimal $O(\sqrt{T})$ Regret and $\widetilde{O}(\sqrt{T})$ CCV by reducing COCO to OCO over surrogate losses. However, their algorithm requires a priori knowledge of the horizon $T$, and its intermediate-time violation bound remains $\widetilde{O}(\sqrt{T})$ rather than $\widetilde{O}(\sqrt{t})$. Follow-up work by Lekeufack & Jordan (2025) extends this reduction to optimistic and dynamic-regret settings, but requires prior bounds on problem-dependent quantities such as path length or prediction error, and therefore relies on the doubling trick for adaptivity. We discuss the doubling trick in more detail in Appendix A.

We also mention recent work on adversarial constraints in partial-information bandit models. Bernasconi et al. (2024) study bandits with stochastic and adversarial constraints and move beyond standard primal–dual methods using optimistic estimates of the feasible set. Bernasconi et al. (2025) study bandits with general long-term constraints through adaptive regret minimization, enforcing weak adaptivity for the primal and dual learners. These works address finite-action bandit feedback, whereas our focus is the full-information OCO setting over continuous convex decision sets. Methodologically, our approach does not update explicit dual variables; instead, it embeds the constraints into time-varying Lyapunov-based surrogate losses, which yields anytime regret and CCV guarantees without restarts.

## 3. Problem Formulation

We consider a repeated game between an online learner and an adaptive adversary. At each round $t \geq 1$, the learner selects an action $x_t$ from a feasible decision set $\mathcal{X}$. The set $\mathcal{X}$ is assumed to be non-empty, closed, and convex, with finite Euclidean diameter $D$. Upon observing the learner's

action $x_t$, the adversary selects two convex and $G$-Lipschitz functions: a *cost function* $f_t : \mathcal{X} \to \mathbb{R}$ and a *constraint function* $g_t : \mathcal{X} \to \mathbb{R}$. The constraint function induces an online constraint of the form $g_t(x) \leq 0$. Since the constraint function $g_t$ is revealed only after the learner chooses $x_t$, it is generally impossible for the learner to satisfy the constraints at every round. Consequently, the learner incurs a cost $f_t(x_t)$ and an instantaneous constraint violation of $\max\{0, g_t(x_t)\}$. Our objective is to design an online policy that achieves small cumulative cost while ensuring a small cumulative constraint violation (CCV) for all $t \geq 1$.

**Performance Metrics:** The (static) regret of an online policy is defined by comparing its cumulative cost against that of the best fixed feasible action in hindsight. Let

$$\mathcal{X}^\star = \{x \in \mathcal{X} : g_t(x) \leq 0, \ \forall t \geq 1\} \qquad (1)$$

denote the set of actions that satisfy all constraints. We assume that the feasible set is non-empty, *i.e.*, $\mathcal{X}^\star \neq \emptyset$. The Regret and cumulative constraint violation (CCV) after $T$ rounds of interactions are defined as

$$\text{Regret}_T = \sup_{x^\star \in \mathcal{X}^\star} \sum_{t=1}^{T} \big(f_t(x_t) - f_t(x^\star)\big), \qquad (2)$$

$$\text{CCV}_T = \sum_{t=1}^{T} \big(g_t(x_t)\big)^+, \qquad (3)$$

where $(y)^+ \triangleq \max\{0, y\}$ for any $y \in \mathbb{R}$.

The classical Online Convex Optimization (OCO) problem is recovered as a special case of COCO in which no constraints are present, *i.e.*, $g_t \equiv 0$ for all $t$, so that $\mathcal{X}^\star = \mathcal{X}$. In this case, the objective reduces to minimizing regret alone (Hazan, 2022).

## 4. Anytime Bounds on Regret and Cumulative Constraint Violation

### 4.1. Technical Overview

An online algorithm is said to be *anytime* if it does not require prior knowledge of the horizon length $T$. Our approach builds on the Lyapunov-based framework recently proposed by Sinha & Vaze (2024), which assumes that $T$ is known in advance. Their method reduces the COCO problem to a standard OCO problem by constructing a sequence of surrogate loss functions that linearly combine the cost and constraint functions. The coefficients of this linear combination are determined by a fixed Lyapunov function of the form

$$\Phi(x) := e^{\lambda x} - 1,$$

where the parameter $\lambda = \frac{1}{2\sqrt{T}}$ depends explicitly on the horizon length. Using this construction, Sinha & Vaze

(2024) show that the resulting algorithm achieves $\text{Regret}_t = O(\sqrt{t})$ and $\text{CCV}_t = O(\sqrt{T})$ for all $t \in [T]$. Consequently, the algorithm is not anytime, and the CCV bound is not time-uniform.

To design an anytime algorithm, we instead employ a sequence of time-varying Lyapunov functions of the form

$$\Phi_t(x) = e^{\lambda_t x} - 1,$$

where the parameter $\lambda_t$ is chosen to be a decreasing function of time. While this removes the dependence on the horizon length, it introduces a significant analytical challenge: the sequence $\Phi_t(Q(t))$ is no longer guaranteed to be monotonically non-decreasing, a property that was central to the analysis of Sinha & Vaze (2024). In particular, the loss of monotonicity breaks a key tensorization argument used in their regret decomposition.

Our main technical contribution is to recover a suitable form of monotonicity by modifying the definition of the constraint accumulation variable. Instead of using the standard additive update, we introduce a multiplicative correction and define

$$Q(t) = \frac{\lambda_{t-1}}{\lambda_t} Q(t-1) + O\big(g_t(x_t)^+\big).$$

Since the sequence $\{\lambda_t\}_{t \geq 1}$ is decreasing, the factor $\frac{\lambda_{t-1}}{\lambda_t} \geq 1$ compensates for the shrinking Lyapunov function, ensuring that

$$\Phi_t(Q(t)) \geq \Phi_{t-1}(Q(t-1)).$$

Moreover, this construction guarantees that $Q(t)$ remains a valid upper bound on the cumulative constraint violation for all $t \geq 1$.

Finally, we construct a surrogate cost function based on the time-varying Lyapunov framework and pass it to a standard adaptive OCO subroutine, such as AdaGrad. As shown later, this approach yields the desired anytime guarantees of $O(\sqrt{t})$ regret and $\tilde{O}(\sqrt{t})$ cumulative constraint violation.

### 4.2. Preliminaries: Lipschitz-Adaptive OCO Algorithms

---

**Algorithm 1** AdaGrad: Online Gradient Descent with Adaptive step sizes

1: **Input** : Convex decision set $\mathcal{X}$, sequence of convex cost functions $\{\hat{f}_t\}_{t=1}^{T}$, sequence of learning rates $\{\eta_t\}_{t=1}^{T}$, $\text{diam}(\mathcal{X}) = D$, $\mathcal{P}_\mathcal{X}(\cdot) = $ Euclidean projection operator on the convex set $\mathcal{X}$.
2: **Initialize** : $x_1 \in \mathcal{X}$ arbitrarily.
3: **for** $t = 1, 2, \ldots$ **do**
4:     Play $x_t$ and compute $\nabla_t = \nabla \hat{f}_t(x_t)$
5:     Set $x_{t+1} = \mathcal{P}_\mathcal{X}(x_t - \eta_t \nabla_t)$
6: **end for**

---

In this section, we briefly review a class of first-order methods, commonly referred to as *Online Gradient Descent (OGD)*, for the standard Online Convex Optimization (OCO) problem. These methods will serve as subroutines in our proposed COCO algorithm. The main distinction among such algorithms lies in the choice of step sizes.

Let $\{\hat{f}_t\}_{t \geq 1}$ be a sequence of convex cost functions. OGD updates its decisions according to (Orabona, 2019, Algorithm 2.1)

$$x_{t+1} = \mathcal{P}_{\mathcal{X}}(x_t - \eta_t \nabla_t), \quad \forall t \geq 1, \tag{4}$$

where $\nabla_t \in \partial \hat{f}_t(x_t)$ is a subgradient of $\hat{f}_t$ at $x_t$, $\mathcal{P}_{\mathcal{X}}(\cdot)$ denotes the Euclidean projection onto the convex set $\mathcal{X}$, and $\{\eta_t\}_{t \geq 1}$ is an adaptive step-size sequence.

A prominent example is the (diagonal) AdaGrad algorithm, which chooses the step size as

$$\eta_t = \frac{\sqrt{2}D}{2\sqrt{\sum_{\tau=1}^{t} G_\tau^2}}, \quad \text{where } G_t = \|\nabla_t\|_2, \ t \geq 1, \tag{5}$$

as proposed in Duchi et al. (2011). (We set $\eta_t = 0$ when $G_t = 0$.)

Importantly, the AdaGrad algorithm does not require knowledge of the horizon length $T$ or a uniform bound on the Lipschitz constants of the loss functions. It enjoys the following adaptive regret guarantee.

**Theorem 1** (Orabona (2019)). *The* AdaGrad *algorithm with step-size sequence given by Eqn.* (5) *achieves the following regret bound for the standard OCO problem:*

$$\text{Regret}_T \leq \sqrt{2}D\sqrt{\sum_{t=1}^{T} G_t^2}. \tag{6}$$

### 4.3. Design and Analysis of an Anytime Algorithm

To simplify the analysis, we pre-process the cost and constraint functions as follows.

On every round, we clip the negative part of the constraint to zero by passing it through the standard ReLU unit. Next, we scale both the cost and constraint functions by a factor of $\alpha \equiv (2GD)^{-1}$. More precisely, we define $\tilde{f}_t \leftarrow \alpha f_t, \tilde{g}_t \leftarrow \alpha(g_t)^+$. Hence, the pre-processed functions are $\alpha G = (2D)^{-1}$-Lipschitz with $\tilde{g}_t \geq 0, \forall t$. In the following, we derive the Regret and CCV bounds for the pre-processed functions. The corresponding bounds for the original functions are obtained upon multiplying the bounds by $\alpha^{-1}$.

#### 4.3.1. SURROGATE COST FUNCTIONS

Let $\{\lambda_t\}_{t \geq 1}$ be a monotonically decreasing sequence to be specified later. We define $Q(t)$ using the following recursion:

$$Q(t) = \frac{\lambda_{t-1}}{\lambda_t}Q(t-1) + \tilde{g}_t(x_t), \ t \geq 1, \tag{7}$$

with $Q(0) = 0$. Since $\frac{\lambda_{t-1}}{\lambda_t} \geq 1$, $Q(t)$ is an upper bound to the $\text{CCV}_t \equiv \sum_{\tau=1}^{t} \tilde{g}_\tau(x_\tau)$ up to round $t$. Thus, an upper bound to $Q(t)$ also implies an upper bound to CCV. Consequently, we proceed to control the sequence $\{Q(t)\}_{t \geq 1}$.

Towards this, let $\Phi_t : \mathbb{R}_+ \to \mathbb{R}_+$ be a sequence of non-decreasing convex Lyapunov functions such that $\Phi_0(0) = 0$. Using the convexity of the function $\Phi_t(\cdot)$, we have

$$\Phi_t(Q(t)) \leq \Phi_t\left(\frac{\lambda_{t-1}}{\lambda_t}Q(t-1)\right) + \Phi_t'(Q(t))\tilde{g}_t(x_t). \tag{8}$$

We now choose the Lyapunov function to be $\Phi_t(x) \overset{\text{(def)}}{=} e^{\lambda_t x} - 1$. Hence, we have $\Phi_t(\frac{\lambda_{t-1}}{\lambda_t}x) = \Phi_{t-1}(x), \forall x$. Substituting this in the bound (8), the one-step change (*drift*) of the potential function $\Phi_t(Q(t))$ can be upper bounded as

$$\Phi_t(Q(t)) - \Phi_{t-1}(Q(t-1)) \leq \Phi_t'(Q(t))\tilde{g}_t(x_t). \tag{9}$$

Recall that, in addition to controlling the CCV, we also want to minimize the cumulative cost (which is equivalent to minimizing Regret). We combine these two objectives into a single objective of minimizing a sequence of surrogate cost functions $\{\hat{f}_t\}_{t=1}^{T}$ obtained by taking a positive linear combination of the drift upper bound (9) and the cost function $\tilde{f}_t$. More precisely, we define the surrogate cost function $\hat{f}_t$ at round $t$ as

$$\hat{f}_t(x) := \tilde{f}_t(x) + \Phi_t'(Q(t))\tilde{g}_t(x), \ t \geq 1. \tag{10}$$

Our proposed anytime policy for COCO, described in Algorithm 2, simply runs the AdaGrad algorithm on the surrogate cost function sequence $\{\hat{f}_t\}_{t \geq 1}$, for a specific choice of the parameter sequence $\{\lambda_t\}_{t \geq 1}$.

#### 4.3.2. THE REGRET DECOMPOSITION INEQUALITY

Let $x^\star \in \mathcal{X}^\star$ be any fixed feasible benchmark action as given by Eqn. (1). Using the drift inequality from Eqn. (9), the definition of surrogate costs (10), and the fact that $g_\tau(x^\star) \leq 0, \forall \tau \geq 1$, we have

$$\Phi_\tau(Q(\tau)) - \Phi_{\tau-1}(Q(\tau-1)) + (\tilde{f}_\tau(x_\tau) - \tilde{f}_\tau(x^\star))$$
$$\leq \hat{f}_\tau(x_\tau) - \hat{f}_\tau(x^\star), \ \forall \tau \geq 1.$$

Summing up the above inequalities for $1 \leq \tau \leq t$, and using the fact that $\Phi_0(0) = 0$, we obtain

$$\Phi_t(Q(t)) + \text{Regret}_t(x^\star) \leq \text{Regret}_t'(x^\star), \ \forall x^\star \in \mathcal{X}^\star, \tag{11}$$

where $\text{Regret}_t$ on the LHS and $\text{Regret}_t'$ on the RHS of (11) refers to the Regret for learning the pre-processed cost functions $\{\tilde{f}_t\}_{t \geq 1}$ and the surrogate cost functions $\{\hat{f}_t\}_{t \geq 1}$ respectively (see Eqn. (2)). From Eqn. (6), the regret bound

---

**Algorithm 2** Anytime Online Policy for COCO

---

1: **Input:** Stream of convex cost functions $\{f_t\}_{t\geq 1}$ and constraint functions $\{g_t\}_{t\geq 1}$, an upper bound $G$ on their subgradient norms, and the diameter $D$ of $\mathcal{X}$.

2: **Parameters:** $\lambda_t = \frac{1}{4\sqrt{t}\,\sqrt{\log t+1}\,(\log(\log t+1)+1)}, \Phi_t(x) = \exp(\lambda_t x) - 1, \alpha = (2GD)^{-1}$, with $\lambda_0 = \lambda_1$.

3: **Initialization:** Set $x_1 \in \mathcal{X}$ arbitrarily and $Q(0) = 0$.

4: **for** $t = 1, 2, \ldots$ **do**

5:    Select action $x_t$, observe $f_t, g_t$, incur a cost of $f_t(x_t)$ and constraint violation of $(g_t(x_t))^+$

6:    $\tilde{f}_t \leftarrow \alpha f_t, \tilde{g}_t \leftarrow \alpha \max(0, g_t)$.

7:    $Q(t) = \frac{\lambda_{t-1}}{\lambda_t} Q(t-1) + \tilde{g}_t(x_t)$.

8:    Compute $\hat{f}_t$ as defined in (10)

9:    Pass $\hat{f}_t$ and $\eta_t = \frac{\sqrt{2}D}{2\sqrt{\sum_{\tau=1}^{t} ||\nabla_\tau||_2^2}}$ to Algorithm 1.

10: **end for**

---

for the AdaGrad algorithm depends on the $\ell_2$ norms of the gradients of the input cost functions. Since we use the AdaGrad subroutine for learning the surrogate cost functions $\{\hat{f}_t\}_{t\geq 1}$, we need to upper bound the gradients of the surrogate functions to derive the regret expression. Towards this, the $\ell_2$-norm of the gradients $G_t$ of the surrogate cost function $\hat{f}_t$ can be bounded using the triangle inequality as follows:

$$
\begin{aligned}
G_t &\leq ||\nabla \tilde{f}_t(x_t)||_2 + \Phi_t'(Q(t))||\nabla \tilde{g}_t(x_t)||_2 \\
&\leq (2D)^{-1}\big(1 + \Phi_t'(Q(t))\big).
\end{aligned}
\tag{12}
$$

where in the last step, we have used the fact that the pre-processed functions are $(2D)^{-1}$-Lipschitz. Hence, plugging in the adaptive regret bound (6) on the RHS of (11), we arrive at the following regret decomposition inequality valid for any $t \geq 1$:

$$
\Phi_t(Q(t)) + \mathsf{Regret}_t(x^\star) \leq \sqrt{t} + \sqrt{\sum_{\tau=1}^{t}\big(\Phi_\tau'(Q(\tau))\big)^2}. \tag{13}
$$

In the above, we have utilized simple algebraic inequalities $(x + y)^2 \leq 2(x^2 + y^2)$ and $\sqrt{a+b} \leq \sqrt{a} + \sqrt{b}, a, b \geq 0$. Also note that, for our choice of Lyapunov function, $\Phi_\tau'(x) = \lambda_\tau e^{\lambda_\tau x} = \lambda_\tau(\Phi_\tau(x) + 1)$. Hence,

$$
\sqrt{\sum_{\tau=1}^{t}\big(\Phi_\tau'(Q(\tau))\big)^2} \leq \sqrt{\sum_{\tau=1}^{t}\lambda_\tau^2\big(\Phi_\tau(Q(\tau)) + 1\big)^2}. \tag{14}
$$

Further note that the recursion $Q(\tau) = \frac{\lambda_{\tau-1}}{\lambda_\tau}Q(\tau - 1) + \tilde{g}_\tau(x_\tau)$ implies $\lambda_\tau Q(\tau) \geq \lambda_{\tau-1}Q(\tau - 1)$ as $\tilde{g}_\tau \geq 0$. Therefore, $\Phi_\tau(Q(\tau)) \geq \Phi_{\tau-1}(Q(\tau - 1))$, *i.e.*, the sequence $\{\Phi_\tau(Q(\tau))\}_{\tau\geq 1}$ is non-negative and non-decreasing. Hence, upper-bounding all terms within the

parentheses in the summation of the RHS of (14) by the last term, we arrive at the following simplified bound

$$
\sqrt{\sum_{\tau=1}^{t}\lambda_\tau^2\big(\Phi_\tau(Q(\tau)) + 1\big)^2} \leq \sqrt{\sum_{\tau=1}^{t}\lambda_\tau^2\big(\Phi_t(Q(t)) + 1\big)}.
$$

Thus Eqn. (13) yields for any $t \geq 1$:

$$
\begin{aligned}
&\Phi_t(Q(t)) + \mathsf{Regret}_t(x^\star) \\
&\leq \sqrt{\sum_{\tau=1}^{t}\lambda_\tau^2 \Phi_t(Q(t))} + \sqrt{\sum_{\tau=1}^{t}\lambda_\tau^2} + \sqrt{t}. \tag{15}
\end{aligned}
$$

The regret decomposition inequality (15) constitutes the key step for the subsequent analysis. We finally choose the parameters

$$
\lambda_\tau = \frac{1}{4\sqrt{\tau}\,\sqrt{\log(\tau) + 1}\,(\log(\log(\tau) + 1) + 1)}, \tau \geq 1.
$$

Note that we have $\sum_{\tau=1}^{t}\lambda_\tau^2 < \frac{1}{4}$, which follows from simple calculus outlined in Lemma 9. Hence, from (15), we conclude that

$$
\mathsf{Regret}_t(x^\star) + \frac{1}{2}\Phi_t(Q(t)) \leq \sqrt{t} + \frac{1}{2}. \tag{16}
$$

Noting that, $\Phi_t(Q(t)) \geq 0$, the above inequality implies the following regret upper bound valid for any $t \geq 1$:

$$
\mathsf{Regret}_t(x^\star) \leq \sqrt{t} + 1.
$$

To upper bound the CCV, note that we trivially have $\mathsf{Regret}_t(x^\star) \geq -\frac{Dt}{2D} \geq -\frac{t}{2}$. Hence, Eqn. (16) implies that

$$
\begin{aligned}
\Phi_t(Q(t)) &\leq t + 2\sqrt{t} + 1, \\
\exp(\lambda_t Q(t)) = \Phi_t(Q(t)) + 1 &\leq t + 2\sqrt{t} + 2 \leq 5t, \\
Q(t) &\leq \frac{1}{\lambda_t}\log(5t) \\
&\leq 4\sqrt{t}\,\sqrt{\log t+1}\,\big(\log(\log t+1) + 1\big)\log(5t).
\end{aligned}
$$

Further, noted above, since $\{\lambda_t\}_{t\in[T]}$ is a decreasing sequence, $\mathsf{CCV}_t \leq Q(t)$ based on the definition of the $\{Q(t)\}$ sequence. We summarize the above results in the following theorem:

**Theorem 2.** *Consider the COCO problem with adversarially chosen $G$-Lipschitz cost and constraint functions and a decision set of diameter $D$. In this setting, Algorithm 2 yields the following* $\mathsf{Regret}$ *and* $\mathsf{CCV}$ *bounds for any $t \geq 1$:*

$$
\begin{aligned}
\mathsf{Regret}_t &\leq 2GD(\sqrt{t} + 1), \\
\mathsf{CCV}_t &\leq 8GD\sqrt{t}\,\sqrt{\log t+1} \\
&\quad \times \big(\log(\log t+1) + 1\big)\log(5t).
\end{aligned}
$$

**Remark 3** (Order optimality). *The regret bound in Theorem 2 matches the minimax $\Omega(\sqrt{t})$ lower bound for standard OCO. Moreover, Sinha & Vaze (2024, Theorem 3) establish a known-horizon COCO lower bound showing that regret and cumulative constraint violation cannot both be $o(\sqrt{T})$ in general. Since Theorem 2 holds simultaneously for every prefix $t \geq 1$, the anytime guarantees are order-optimal up to logarithmic factors at every time $t$.*

**Remark 4** (Adaptive variant under first-order feedback). *The dependence on a known global Lipschitz constant $G$ can be removed under the standard first-order OCO feedback model, where after playing $x_t$ the learner observes local subgradients of the revealed functions. Appendix C gives a variant whose OCO step sizes and Lyapunov parameters depend on observed gradient norms, yielding data-dependent bounds of order*

$$O\left(\sqrt{\sum_{\tau=1}^{t} \|\nabla f_\tau(x_\tau)\|_2^2}\right) \quad and$$

$$\widetilde{O}\left(\sqrt{\sum_{\tau=1}^{t} \|\nabla g_\tau^+(x_\tau)\|_2^2}\right)$$

*for regret and* CCV, *respectively. Thus this variant does not require an a priori uniform upper bound on $G$ or knowledge of $T$, while still assuming standard post-decision first-order feedback and knowledge of the ambient diameter $D = \text{diam}(\mathcal{X})$.*

## 5. Extension to Dynamic Regret Bounds

While in the previous section, Algorithm 2 ensured a static regret guarantee against a fixed benchmark, in this section, we show that the same algorithm also yields a dynamic regret guarantee against any sequence of time-varying benchmarks. More importantly, our algorithm does not need to know any upper bound on the worst-case path length corresponding to the benchmark sequence.

### 5.1. Adaptive OCO Meta-Algorithm to Minimize Dynamic Regret

Let $\{h_t\}_{t\geq 1}$ be a generic sequence of convex OCO losses and let $\{x_t^\star\}_{t\geq 1}$ be a comparator sequence. In the constrained application below, the comparators satisfy $g_t(x_t^\star) \leq 0$ at every round. The dynamic regret against $\{x_t^\star\}_{t=1}^{T}$ of a policy producing $\{x_t\}_{t\geq 1}$ is

$$\text{D-Regret}_T \equiv \sum_{t=1}^{T} h_t(x_t) - \sum_{t=1}^{T} h_t(x_t^\star). \quad (17)$$

---

**Algorithm 3** Anytime Online Policy for Dynamic COCO

1: **Input:** Stream of convex cost functions $\{f_t\}_{t\geq 1}$ and constraint functions $\{g_t\}_{t\geq 1}$, an upper bound $G$ on their subgradient norms, and the diameter $D$ of $\mathcal{X}$.

2: **Parameters:** $\alpha = (2GD)^{-1}$ and $\lambda_t = \frac{1}{4\sqrt{t(1+\mathcal{P}_t)}\sqrt{\log t+1}\,(\log(\log t+1)+1)}$, with $\lambda_0 = \lambda_1$.

3: **Initialization:** Set $x_1 \in \mathcal{X}$ arbitrarily and $Q(0) = 0$.

4: **for** $t = 1, 2, \ldots$ **do**

5:     Choose $x_t$, observe $f_t, g_t$, incur a cost of $f_t(x_t)$ and constraint violation of $(g_t(x_t))^+$

6:     $\tilde{f}_t \leftarrow \alpha f_t, \tilde{g}_t \leftarrow \alpha \max(0, g_t)$.

7:     $Q(t) = \frac{\lambda_{t-1}}{\lambda_t} Q(t-1) + \tilde{g}_t(x_t)$.

8:     Compute $\hat{f}_t$ as per (10)

9:     Pass $\hat{f}_t$ and $\eta_t = \frac{(D+1)(1+\mathcal{P}_t)}{\sqrt{2\sum_{\tau=1}^{t}(1+\mathcal{P}_\tau)\|\nabla_\tau\|^2}}$ to Algorithm 1.

10: **end for**

---

We define the path length $\mathcal{P}_T$ of a comparator sequence $\{x_t^\star\}_{t=1}^{T}$ as follows:

$$\mathcal{P}_T \equiv \sum_{t=1}^{T-1} \|x_{t+1}^\star - x_t^\star\|_2. \quad (18)$$

Note that for the static regret metric defined in (2), we have $x_t^\star = x^\star, \forall t \geq 1$, and hence, the path-length is zero (Hazan, 2022). For any round $t \geq 1$, let $x_t^\star$ be an optimal feasible action, *i.e.,* it is a solution to the following constrained convex optimization problem:

$$\min f_t(x), \quad \text{s.t. } g_t(x) \leq 0, \quad x \in \mathcal{X}. \quad (19)$$

By definition, the worst-case dynamic regret upper bounds the dynamic regret for any arbitrary feasible comparator sequence. Also, the advantage of considering the *worst-case* benchmarks is that at the end of each round $t$, we can compute the benchmark $x_t^\star$ by solving problem (19). This enables us to determine the running value of $\mathcal{P}_t$ at the end of each round $t$.

For the time being, let us now consider the standard OCO problem without any constraints. The following theorem gives an upper bound to the dynamic regret of the adaptive OGD policy described in Algorithm 1 for the standard OCO problem.

**Theorem 5** (Dynamic Regret Bounds for AdaGrad). *The* AdaGrad *policy, described in Algorithm 1, achieves the following adaptive dynamic regret bound for the standard OCO problem against any comparator sequence $u_{1:T}$ whose path length at time $t$ is known to be $\mathcal{P}_t$, using an adaptive*

*learning rate schedule* $\eta_t = \frac{(D+1)(1+\mathcal{P}_t)}{\sqrt{2\sum_{\tau=1}^{t}(1+\mathcal{P}_\tau)\|\nabla_\tau\|^2}}, t \geq 1:$

$$\text{D-Regret}_T(\mathcal{P}_T) \leq (D+1)\sqrt{2\sum_{t=1}^{T}(1+\mathcal{P}_t)\|\nabla_t\|^2}.$$
$$(20)$$

*where* $\nabla_t \equiv \nabla h_t(x_t), \forall t \geq 1$ *and* $diam(\mathcal{X}) = D$.

**Remark 6.** *Theorem 5 is used as a base OCO ingredient for the constrained-to-unconstrained reduction, rather than as the main standalone contribution. Its role is to provide a dynamic-regret guarantee whose learning rate depends only on the path length observed up to time t, which can be computed online for the worst-case comparator sequence. This is what allows the Lyapunov-based COCO reduction to obtain anytime dynamic-regret and* CCV *bounds without knowing the terminal path length* $\mathcal{P}_T$ *in advance. The proof adapts the static regret analysis of* AdaGrad *from Orabona (2019, Theorem 4.14); see Appendix D. Recent work by Supantha & Sinha (2025, Theorem 17) gives a related dynamic-regret bound, but assumes prior knowledge of the total path length* $\mathcal{P}_T$.

### 5.2. Dynamic Regret Decomposition Inequality

We use the same surrogate loss as defined earlier in Eqn. (10). This yields

$$\Phi_\tau(Q(\tau)) - \Phi_{\tau-1}(Q(\tau-1)) + \tilde{f}_\tau(x_\tau) - \tilde{f}_\tau(x_\tau^\star)$$
$$\overset{(a)}{\leq} \Phi_\tau'(Q(\tau))\big(\tilde{g}_\tau(x_\tau) - \tilde{g}_\tau(x_\tau^\star)\big) + \tilde{f}_\tau(x_\tau) - \tilde{f}_\tau(x_\tau^\star)$$
$$\overset{(b)}{=} \hat{f}_\tau(x_\tau) - \hat{f}_\tau(x_\tau^\star)$$

where in step (a), we have used Eqn. (9) and the feasibility of the benchmark $x_\tau^\star$ (which implies $g_\tau(x_\tau^\star) \leq 0$), and in step (b) we have used the definition of surrogate costs from Eqn. (10). Summing up the above inequalities for $\tau = 1$ to $\tau = t$, we have the following dynamic regret decomposition inequality

$$\Phi_t(Q(t)) + \text{D-Regret}_t(\mathcal{P}_t) \leq \text{D-Regret}_t'(\mathcal{P}_t), \quad (21)$$

where $\text{D-Regret}_t(\mathcal{P}_t)$ and $\text{D-Regret}_t'(\mathcal{P}_t)$ correspond to the dynamic regrets up to round $t$ for the original cost functions $\{\tilde{f}_\tau\}_{\tau\geq 1}$ and surrogate cost functions $\{\hat{f}_\tau\}_{\tau\geq 1}$ respectively against a sequence of benchmarks with path length at most $\mathcal{P}_t$. Plugging in the dynamic regret expression for the surrogate costs from Theorem 5, we conclude that:

$$\Phi_t(Q(t)) + \text{D-Regret}_t(\mathcal{P}_t)$$
$$\leq \sqrt{1+\mathcal{P}_t}\sqrt{t} + \sqrt{\sum_{\tau=1}^{t}(1+\mathcal{P}_\tau)\big(\Phi_\tau'(Q(\tau))\big)^2}. \quad (22)$$

The rest of the analysis is similar to that given in Section 4.3.2 by taking $\Phi_\tau(x) = e^{\lambda_\tau x} - 1$. We now choose

$\lambda_\tau = \frac{1}{4\sqrt{\tau(1+\mathcal{P}_\tau)}\sqrt{\log(\tau)+1}(\log(\log(\tau)+1)+1)}$. We formally state our results in the following theorem,

**Theorem 7.** *For the COCO problem with adversarially chosen G-Lipschitz cost and constraint functions, Algorithm 3 yields the following worst-case dynamic regret and* CCV *bounds for any* $t \geq 1$

$$\text{D-Regret}_t(\mathcal{P}_t) \leq 2GD\sqrt{1+\mathcal{P}_t}(\sqrt{t}+1),$$
$$\text{CCV}_t \leq 8GD\sqrt{1+\mathcal{P}_t}\sqrt{t}\sqrt{\log t + 1}$$
$$\times \big(\log(\log t + 1)+1\big)\log(5t),$$

*where* $\mathcal{P}_t$ *is the path length of the worst-case comparator sequence* $\{x_\tau^\star\}_{\tau=1}^{t}$ *up to time t.*

## 6. An Adaptive Optimistic Algorithm

Generalizing the seminal work of Sinha & Vaze (2024), Lekeufack & Jordan (2025) presents an optimistic meta-algorithm for Constrained Online Convex Optimization (COCO) that leverages predictions of the loss and constraint functions to achieve superior performance when those predictions are accurate. However, their algorithm requires tuning a crucial parameter that depends on the total prediction error over the entire horizon - a quantity that is unknown in advance. Consequently, to make their algorithm adaptive, they resort to the doubling trick. In this section, we propose a modified algorithm that is both optimistic and continuously adaptive, obviating the use of the doubling trick.

### 6.1. Optimistic Meta-Algorithm

---
**Algorithm 4** Optimistic COCO

1: **Input:** Stream of convex cost functions $\{f_t\}_{t\geq 1}$ and constraint functions $\{g_t\}_{t\geq 1}$, an upper bound $G$ on their subgradient norms, and the diameter $D$ of $\mathcal{X}$.
2: **Initialization:** Set $x_1 \in \mathcal{X}$ arbitrarily and $Q(0) = Q(1) = 0$.
3: **for** round $t = 1, 2, \ldots$ **do**
4:     Choose $x_t$, observe $f_t, g_t$, incur a cost of $f_t(x_t)$ and constraint violation of $(g_t(x_t))^+$
5:     $\tilde{f}_t \leftarrow \alpha f_t, \tilde{g}_t \leftarrow \alpha \max(0, g_t)$.
6:     Predict $\bar{f}_{t+1}$ and $\bar{g}_{t+1}$.
7:     Compute $\hat{f}_t$ as per (27).
8:     Update $Q(t+1) = \frac{\lambda_t}{\lambda_{t+1}}Q(t) + \tilde{g}_t(x_t)$.
9:     Compute prediction $\hat{\bar{f}}_{t+1}$ as in (24).
10:    Pass $\hat{f}_t$ and $\hat{\bar{f}}_{t+1}$ to Algorithm 5.
11: **end for**

---

The above algorithm combines the Optimistic COCO meta-algorithm of Lekeufack & Jordan (2025) with our time-varying Lyapunov functions and modified queue recursion.

It constructs the true and predicted surrogate losses ($\hat{f}_t$ and $\hat{\bar{f}}_{t+1}$) using the adaptive queue $Q(t)$ and passes them to Optimistic Online Mirror Descent (Algorithm 5). The base regret guarantee is stated in Theorem 10; the algorithm originates in Rakhlin & Sridharan (2014). We next define the prediction errors and the predicted surrogate loss, state the regret and constraint-violation guarantee, and defer the proof to Appendix E.2.

In the optimistic setting, we assume that at the end of step $t$, the learner can make predictions $\bar{f}_{t+1}$ and $\bar{g}_{t+1}$. More precisely, we are interested in predictions of the gradients, and, for any function $h$, we denote by $\nabla \bar{h}_t$ the prediction of the gradient of $h$. We denote by $\bar{h}_t$ the function whose gradient is $\nabla \bar{h}_t$. Moreover, we define the following prediction errors

$$
\begin{aligned}
\epsilon_t(h) &:= \|\nabla h_t(x_t) - \nabla \bar{h}_t(x_t)\|_*^2, \\
\mathcal{E}_t(h) &:= \sum_{\tau=1}^{t} \epsilon_\tau(h).
\end{aligned} \tag{23}
$$

In the previous algorithms, we formed $\hat{f}_t$ at the end of round $t$ using the revealed cost $f_t$ and constraint $g_t$. Here, we will also form the predicted surrogate cost $\hat{\bar{f}}_{t+1}$ using the predicted cost $\bar{f}_{t+1}$ and constraint $\bar{g}_{t+1}$. We assume without loss of generality that both the actual and predicted functions have the same Lipschitz constant, which upon pre-processing is $(2D)^{-1}$. This also implies that both $\epsilon_t(\tilde{f})$ and $\epsilon_t(\tilde{g})$ are bounded from above by $D^{-2}$.

We define the predicted surrogate cost $\hat{\bar{f}}_{t+1}$ as

$$
\hat{\bar{f}}_{t+1} = \bar{f}_{t+1} + \Phi'_{t+1}(Q(t+1))\bar{g}_{t+1} \tag{24}
$$

However, note that if we were to retain the usual definition of $Q(t)$, the above equation would imply that we know the value of $\tilde{g}_{t+1}(x_{t+1})$ at the end of round $t$. This is not possible, therefore, Lekeufack & Jordan (2025) has to use delayed updates, wherein,

$$
Q(t+1) = Q(t) + \tilde{g}_t(x_t) \tag{25}
$$

To apply our adaptive technique we have to augment this with our multiplicative factor to get,

$$
Q(t+1) = \frac{\lambda_t}{\lambda_{t+1}} Q(t) + \tilde{g}_t(x_t) \tag{26}
$$

Note that, this means $\mathsf{CCV}_t \leq Q(t+1)$. Further, $\lambda_{t+1}$ should be such that it is known after round $t$.

Also note that for parity with the definition of predicted surrogate loss, we need to modify our definition of surrogate loss.

$$
\hat{f}_t = \tilde{f}_t + \Phi'_t(Q(t))\tilde{g}_t \tag{27}
$$

where $Q(t)$ follows the delayed multiplicative update above.

We further choose our $\lambda_\tau$ whose value will be justified in subsequent analysis.

$$
\lambda_\tau = \frac{1}{c\sqrt{\gamma_\tau}\sqrt{\log(\gamma_\tau) + 1}(\log(\log(\gamma_\tau) + 1) + 1)}. \tag{28}
$$

where

$$
\gamma_\tau = \mathcal{E}_{\tau-1}(\tilde{g}) + D^{-2} + 1, \ c = 20\left(\sqrt{\frac{B}{\beta}} + \frac{B}{\beta}\right). \tag{29}
$$

We summarize our results in the following theorem.

**Theorem 8.** *For the COCO problem with adversarially chosen $G$-Lipschitz cost and constraint functions, Algorithm 4, using Algorithm 5 as its base OCO algorithm, yields the following Regret and $\mathsf{CCV}$ bounds for any $t \geq 1$:*

$$
Regret_t \leq O\left(\sqrt{\mathcal{E}_t(\tilde{f})}\right), \ \mathsf{CCV}_t \leq \tilde{O}\left(\sqrt{\mathcal{E}_t(\tilde{g})}\right).
$$

For completeness, we provide the proof of the above theorem in Appendix E.2.

## 7. Simulations

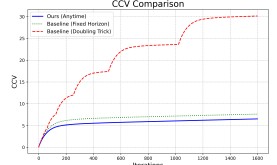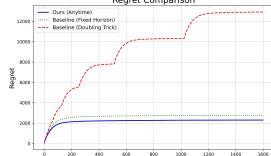

*Figure 1.* Online shortest-path experiment. Left: cumulative constraint violation. Right: cumulative regret. Our anytime policy avoids the restart-induced instability of the doubling trick and performs competitively with, and in this experiment better than, the fixed-horizon baseline.

We evaluate Algorithm 2 on a constrained online shortest-path problem with time-varying latency costs and bandwidth constraints. We compare against the fixed-horizon algorithm of Sinha & Vaze (2024) and its standard doubling-trick adaptation. Figure 1 shows that the proposed anytime method avoids the periodic instability caused by restarts and achieves lower cumulative regret and CCV in this experiment. The full graph construction, data generation procedure, and additional discussion are deferred to Appendix F.

## 8. Conclusion

In this work, we proposed anytime algorithms for online convex optimization with adversarial constraints (COCO), covering static regret, dynamic regret, and optimistic settings.

Our approach employs time-varying Lyapunov functions together with a multiplicative correction in the virtual-queue recursion, which restores the monotonicity needed for the regret–violation decomposition while avoiding doubling-trick restarts. The resulting algorithms are more stable in practice, as illustrated by the constrained online shortest-path experiment.

**Limitations and Open Problems.** Our main anytime guarantees remove the need to know the horizon in advance, and the adaptive variant in Appendix C removes the need for an a priori uniform Lipschitz bound under standard first-order feedback. However, the algorithms still rely on knowledge of the ambient diameter $D = \mathrm{diam}(\mathcal{X})$. In particular, the constants scale with $D$, rather than with the potentially smaller diameter $D^\star = \mathrm{diam}(\mathcal{X}^\star)$ of the hindsight-feasible set, because future adversarial constraints are unknown and the learner cannot project onto $\mathcal{X}^\star$ online. Designing fully parameter-free anytime COCO algorithms, or algorithms whose constants depend only on $D^\star$, remains an interesting open problem. Another open direction is to obtain anytime regret–violation trade-offs that reduce CCV at the expense of larger regret.

## Acknowledgements

This work was supported by the Department of Atomic Energy, Government of India, under project no. RTI4014 and by a Google India faculty research award.

## Impact Statement

This paper presents work aimed at advancing the field of machine learning. There are many potential societal consequences of our work, none of which we feel must be specifically highlighted here.

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

## A. The Doubling Trick

The doubling trick is a standard technique for adapting online algorithms that require a known time horizon $T$ to a setting where the horizon is unknown. The method works by running the algorithm in exponentially increasing phases. At the start of each new phase, the algorithm is restarted, and its internal parameters (which depend on $T$) are recalculated using the new, doubled phase length as the estimate for $T$.

However, as noted in several prior works, this technique, while theoretically functional, is often criticized for being "aesthetically inelegant and impractical" (Luo & Schapire, 2014; Zhang et al., 2024; Kwon & Mertikopoulos, 2014). A major line of critique, articulated by Luo & Schapire (2014), is that the method is "intuitively wasteful, since it repeatedly restarts itself, entirely forgetting all the preceding information". Their work, which proposes an alternative adaptive algorithm based on a "pretend prior distribution" over the horizon, also demonstrated empirically that the doubling trick is "beaten by most of the other algorithms" in practice.

Similarly, Kwon & Mertikopoulos (2014) developed their continuous-time approach specifically to provide a "unified any-time analysis without needing to reboot the algorithm every so often". Their work shows that by using a *time-varying parameter* (e.g., $\eta_t \propto 1/\sqrt{t}$) rather than a fixed one, one can achieve optimal $\mathcal{O}(t^{-1/2})$ regret bounds without resorting to restarts.

More recently, Zhang et al. (2024) also motivated their work by designing an algorithm that "does not employ the impractical doubling trick". They reiterated that restarting "wastes data" and causes "large jumps" in the decision sequence, which can be undesirable, and that it performs "considerably worse" in practice. As a solution, they proposed a "refined discretization argument" from a continuous-time model to preserve adaptivity without restarts.

The consensus from these works is that continuously adaptive algorithms are generally preferable and perform better in practice. Our own experiments in Appendix F and Figure 1 confirm this finding, showing that our intrinsically adaptive method offers distinctive practical performance gains over the standard doubling trick baseline.

## B. Supporting Lemmas

**Lemma 9.** *For*

$$\lambda_\tau = \frac{1}{4\sqrt{\tau}\,\sqrt{\log(\tau)+1}\,\big(\log(\log(\tau)+1)+1\big)},$$

*we have*

$$\sum_{\tau=1}^{t} \lambda_\tau^2 < \tfrac{1}{4}, \quad \forall\, t \geq 1.$$

*Proof.* We observe

$$\lambda_\tau^2 = \frac{1}{16\,\tau\,(\log(\tau)+1)\,\big(\log(\log(\tau)+1)+1\big)^2}.$$

Define

$$a(x) = \frac{1}{16\,x\,(\log(x)+1)\,\big(\log(\log(x)+1)+1\big)^2}, \quad x \geq 1.$$

The function $a(x)$ is positive and eventually decreasing, so we can apply the integral test:

$$\sum_{\tau=1}^{t} \lambda_\tau^2 \;\leq\; a(1) + \int_1^t a(x)\,dx.$$

We compute

$$\int_1^t a(x)\,dx = \frac{1}{16} \int_1^t \frac{dx}{x(\log x + 1)(\log(\log x + 1)+1)^2}.$$

With the substitution $u = \log x + 1$ (so $du = dx/x$), this becomes

$$= \frac{1}{16} \int_1^{\log t + 1} \frac{du}{u(\log u + 1)^2}.$$

Now let $v = \log u + 1$ so that $dv = du/u$. Then

$$= \frac{1}{16} \int_1^{\log(\log t + 1) + 1} \frac{dv}{v^2} = \frac{1}{16} \left( 1 - \frac{1}{\log(\log t + 1) + 1} \right).$$

Hence

$$\sum_{\tau=1}^t \lambda_\tau^2 \leq a(1) + \frac{1}{16} \left( 1 - \frac{1}{\log(\log t + 1) + 1} \right).$$

Explicitly,

$$a(1) = \frac{1}{16 \cdot 1 \cdot (\log 1 + 1)(\log(\log 1 + 1) + 1)^2} = \frac{1}{16}.$$

Therefore,

$$\sum_{\tau=1}^t \lambda_\tau^2 \leq \frac{1}{16} + \frac{1}{16} \cdot 1 = \frac{1}{8} < \frac{1}{4}.$$

The bound holds uniformly for all $t \geq 1$, proving the claim. $\qquad \square$

## C. Lipschitz-Adaptive COCO Algorithm

We start with the general regret decomposition given in Eqn. (11), however, note that, for this analysis we are not preprocessing the functions since we assume that the Lipschitz constant $G$ is unknown.

$$\Phi_t(Q(t)) + \mathsf{Regret}_t(x^\star) \leq \mathsf{Regret}'_t(x^\star), \ \forall x^\star \in \mathcal{X}^\star,$$

Further, note that,

$$G_t \ \leq \ ||\nabla f_t(x_t)||_2 + \Phi'_t(Q(t))||\nabla g_t^+(x_t)||_2.$$

We also note that $\mathsf{Regret}'_t(x^\star)$ can be bounded in the following way using Theorem 1-

$$\mathsf{Regret}'_t(x^\star) \leq \sqrt{2} D \sqrt{\sum_{\tau=1}^t G_\tau^2} \leq \sqrt{2} D \sqrt{\sum_{\tau=1}^t (||\nabla f_\tau(x_\tau)||_2 + \Phi'_\tau(Q(\tau))||\nabla g_\tau^+(x_\tau)||_2)^2} \tag{30}$$

Using the identities $(a + b)^2 \leq 2(a^2 + b^2)$ and $\sqrt{a^2 + b^2} \leq a + b$ we get,

$$\mathsf{Regret}'_t(x^\star) \leq 2D \sqrt{\sum_{\tau=1}^t ||\nabla f_\tau(x_\tau)||_2^2} + 2D \sqrt{\sum_{\tau=1}^t (\Phi'_\tau(Q(\tau))||\nabla g_\tau^+(x_\tau)||_2)^2} \tag{31}$$

Also note that, for our choice of Lyapunov function, $\Phi'_\tau(x) = \lambda_\tau e^{\lambda_\tau x} = \lambda_\tau(\Phi_\tau(x) + 1)$. Hence,

$$\sqrt{\sum_{\tau=1}^t \left( \Phi'_\tau(Q(\tau)) \right)^2} \leq \sqrt{\sum_{\tau=1}^t \lambda_\tau^2 \left( \Phi_\tau(Q(\tau)) + 1 \right)^2}. \tag{32}$$

Further note that the recursion $Q(\tau) = \frac{\lambda_{\tau-1}}{\lambda_\tau} Q(\tau-1) + g_\tau^+(x_\tau)$ implies $\lambda_\tau Q(\tau) \geq \lambda_{\tau-1} Q(\tau-1)$ as $g_\tau^+ \geq 0$. Therefore, $\Phi_\tau(Q(\tau)) \geq \Phi_{\tau-1}(Q(\tau-1))$, *i.e.*, the sequence $\{\Phi_\tau(Q(\tau))\}_{\tau\geq 1}$ is non-negative and non-decreasing. Hence, upper-bounding all terms within the parentheses in the summation of the RHS of (32) by the last term, we arrive at the following simplified bound

$$\sqrt{\sum_{\tau=1}^t \lambda_\tau^2 \big(\Phi_\tau(Q(\tau)) + 1\big)^2} \leq \sqrt{\sum_{\tau=1}^t \lambda_\tau^2 \big(\Phi_t(Q(t)) + 1\big)}.$$

Thus Eqn. (31) yields for any $t \geq 1$:

$$\Phi_t(Q(t)) + \mathsf{Regret}_t(x^\star) \leq 2D\sqrt{\sum_{\tau=1}^t \|\nabla f_\tau(x_\tau)\|_2^2} + 2D\sqrt{\sum_{\tau=1}^t \lambda_\tau^2 \|\nabla g_\tau^+(x_\tau)\|_2^2 \Phi_t(Q(t))} + 2D\sqrt{\sum_{\tau=1}^t \lambda_\tau^2 \|\nabla g_\tau^+(x_\tau)\|_2^2}. \tag{33}$$

We denote $\gamma_t = 1 + \sum_{\tau=1}^t \|\nabla g_\tau^+(x_\tau)\|_2^2$ and take $\lambda_\tau = \frac{1}{8D\sqrt{\gamma_\tau}\,\sqrt{\log(\gamma_\tau)+1}\,(\log(\log(\gamma_\tau)+1)+1)}$ to get,

Now we can upper bound the following sum,

$$\sum_{\tau=1}^t \lambda_\tau^2 \|\nabla g_\tau^+(x_\tau)\|_2^2 \leq \frac{1}{16D^2} \tag{34}$$

using an approach similar to the Lemma 9.

Therefore, we have,

$$\Phi_t(Q(t)) + \mathsf{Regret}_t(x^\star) \leq 2D\sqrt{\sum_{\tau=1}^t \|\nabla f_\tau(x_\tau)\|_2^2} + \frac{1}{2}\Phi_t(Q(t)) + \frac{1}{2}. \tag{35}$$

$$\frac{1}{2}\Phi_t(Q(t)) + \mathsf{Regret}_t(x^\star) \leq 2D\sqrt{\sum_{\tau=1}^t \|\nabla f_\tau(x_\tau)\|_2^2} + \frac{1}{2}. \tag{36}$$

Overall, we get $\mathsf{Regret}_t(x^\star) = 2D\sqrt{\sum_{\tau=1}^t \|\nabla f_\tau(x_\tau)\|_2^2}$ and $\mathsf{CCV}_t = \widetilde{O}\left(D\sqrt{\sum_{\tau=1}^t \|\nabla g_\tau^+(x_\tau)\|_2^2}\right)$

These bounds are tighter because they can adjust themselves based on the problem instance, whereas the $\tilde{O}(\sqrt{t})$ bounds were worst-case bounds which remained the same for all problem instances.

## D. Proof of Theorem 5

Define $y_{t+1} := x_t - \eta_t \nabla \hat{f}_t(x_t)$, where the non-increasing adaptive step size sequence $\{\eta_t\}_{t\geq 1}$ has been defined in Algorithm 1. For any feasible comparator action $x_t^\star \in \mathcal{X}$, we have

$$\|x_{t+1} - x_t^\star\|^2 \overset{(a)}{\leq} \|y_{t+1} - x_t^\star\|^2 = \|x_t - x_t^\star\|^2 + \eta_t^2\|\nabla_t\|^2 - 2\eta_t \nabla_t^\top(x_t - x_t^\star),$$

where inequality (a) follows from the non-expansive property of Euclidean projection and the second equality follows from the definition of $y_{t+1}$. Rearranging the above inequality, we have:

$$2\nabla_t^\top(x_t - x_t^\star) \leq \frac{\|x_t - x_t^\star\|^2 - \|x_{t+1} - x_t^\star\|^2}{\eta_t} + \eta_t\|\nabla_t\|^2. \tag{37}$$

Using the convexity of the cost functions and summing the above inequalities over $1 \le t \le T$, we obtain:

$$2\sum_{t=1}^{T}\left(\hat{f}_t(x_t) - \hat{f}_t(x_t^\star)\right) \le 2\sum_{t=1}^{T}\nabla_t^\top(x_t - x_t^\star) \le \underbrace{\sum_{t=1}^{T}\frac{||x_t - x_t^\star||^2 - ||x_{t+1} - x_t^\star||^2}{\eta_t}}_{(A)} + \sum_{t=1}^{T}\eta_t||\nabla_t||^2. \tag{38}$$

Next we simplify term (A) in Eqn. (38). It can be expressed as

$$\frac{||x_1^\star - x_1||^2}{\eta_1} - \frac{||x_T^\star - x_{T+1}||^2}{\eta_T} + \sum_{t=1}^{T-1}\frac{||x_{t+1} - x_{t+1}^\star||^2}{\eta_{t+1}} - \frac{||x_{t+1} - x_t^\star||^2}{\eta_t}$$

$$= \frac{||x_1^\star - x_1||^2}{\eta_1} - \frac{||x_T^\star - x_{T+1}||^2}{\eta_T} + \underbrace{\sum_{t=1}^{T-1}\frac{\eta_t||x_{t+1} - x_{t+1}^\star||^2 - \eta_{t+1}||x_{t+1} - x_t^\star||^2}{\eta_t\eta_{t+1}}}_{(B)}. \tag{39}$$

We next upper bound term (B) in (39).

$$\sum_{t=1}^{T-1}\frac{||\sqrt{\eta_t}x_{t+1} - \sqrt{\eta_t}x_{t+1}^\star||^2 - ||\sqrt{\eta_{t+1}}x_{t+1} - \sqrt{\eta_{t+1}}x_t^\star||^2}{\eta_t\eta_{t+1}}$$

$$= \sum_{t=1}^{T-1}\frac{\langle(\sqrt{\eta_t} + \sqrt{\eta_{t+1}})x_{t+1} - \sqrt{\eta_t}x_{t+1}^\star - \sqrt{\eta_{t+1}}x_t^\star, (\sqrt{\eta_t} - \sqrt{\eta_{t+1}})x_{t+1} - \sqrt{\eta_t}x_{t+1}^\star + \sqrt{\eta_{t+1}}x_t^\star\rangle}{\eta_t\eta_{t+1}}$$

$$\le \sum_{t=1}^{T-1}\frac{||(\sqrt{\eta_t} + \sqrt{\eta_{t+1}})x_{t+1} - \sqrt{\eta_t}x_{t+1}^\star - \sqrt{\eta_{t+1}}x_t^\star||\,||(\sqrt{\eta_t} - \sqrt{\eta_{t+1}})x_{t+1} - \sqrt{\eta_t}x_{t+1}^\star + \sqrt{\eta_{t+1}}x_t^\star||}{\eta_t\eta_{t+1}},$$

where the last step follows from an application of the Cauchy-Schwarz inequality. Note that the first term in the numerator can be bounded as:

$$||(\sqrt{\eta_t} + \sqrt{\eta_{t+1}})x_{t+1} - \sqrt{\eta_t}x_{t+1}^\star - \sqrt{\eta_{t+1}}x_t^\star|| \le \sqrt{\eta_t}||x_{t+1} - x_{t+1}^\star|| + \sqrt{\eta_{t+1}}||x_{t+1} - x_t^\star|| \le (\sqrt{\eta_t} + \sqrt{\eta_{t+1}})D,$$

where we have used the triangle inequality and the feasibility of the algorithm's and benchmark's actions in the last step. Using this, we have

$$(B) \le D\sum_{t=1}^{T-1}\frac{(\sqrt{\eta_t} + \sqrt{\eta_{t+1}})\,||(\sqrt{\eta_t} - \sqrt{\eta_{t+1}})x_{t+1} - \sqrt{\eta_t}x_{t+1}^\star + \sqrt{\eta_{t+1}}x_t^\star||}{\eta_t\eta_{t+1}}$$

$$= D\sum_{t=1}^{T-1}\frac{(\sqrt{\eta_t} + \sqrt{\eta_{t+1}})||(\sqrt{\eta_t} - \sqrt{\eta_{t+1}})(x_{t+1} - x_{t+1}^\star) + \sqrt{\eta_{t+1}}(x_t^\star - x_{t+1}^\star)||}{\eta_t\eta_{t+1}}$$

$$\overset{(a)}{\le} D\sum_{t=1}^{T-1}\frac{(\sqrt{\eta_t} + \sqrt{\eta_{t+1}})\,[(\sqrt{\eta_t} - \sqrt{\eta_{t+1}})D + \sqrt{\eta_{t+1}}||x_t^\star - x_{t+1}^\star||]}{\eta_t\eta_{t+1}}$$

$$\overset{(b)}{\le} D\sum_{t=1}^{T-1}\frac{(\eta_t - \eta_{t+1})D + 2\eta_t||x_{t+1}^\star - x_t^\star||}{\eta_t\eta_{t+1}}$$

$$= D^2\Big(\frac{1}{\eta_T} - \frac{1}{\eta_1}\Big) + 2D\sum_{t=1}^{T-1}\frac{||x_{t+1}^\star - x_t^\star||}{\eta_{t+1}}$$

$$\overset{(c)}{\le} D^2\Big(\frac{1}{\eta_T} - \frac{1}{\eta_1}\Big) + \frac{2D\mathcal{P}_T(x_{1:T}^\star)}{\eta_T},$$

where in step (a), we have used the triangle inequality combined with the feasibility of the algorithm's and benchmark's actions, and in step (c), we have used the definition of the path length of the comparator. The non-increasing property of the

step sizes, *i.e.*, $\eta_t \geq \eta_{t+1}, \forall t \geq 1$ was used in steps (a), (b), and (c). Finally, combining the above bound with Eqns. (38) and (39), we conclude

$$2\left(\sum_{t=1}^{T} \hat{f}_t(x_t) - \hat{f}_t(x_t^\star)\right)$$

$$\leq \frac{||x_1^\star - x_1||^2}{\eta_1} - \frac{||x_T^\star - x_{T+1}||^2}{\eta_T} + D^2\left(\frac{1}{\eta_T} - \frac{1}{\eta_1}\right) + \frac{2D\mathcal{P}_T(x_{1:T}^\star)}{\eta_T} + \sum_{t=1}^{T} \eta_t ||\nabla_t||^2$$

$$\leq \frac{D^2 + 2D\mathcal{P}_T(x_{1:T}^\star)}{\eta_T} + \sum_{t=1}^{T} \eta_t ||\nabla_t||^2.$$

Hence, the dynamic regret of Algorithm 1 can be upper bounded as

$$\text{D-Regret}_T \leq \frac{\max(D^2, 2D)(1 + \mathcal{P}_T(x_{1:T}^\star))}{2\eta_T} + \underbrace{\frac{1}{2}\sum_{t=1}^{T} \eta_t ||\nabla_t||^2}_{(C)}. \tag{40}$$

The dynamic regret bound in Eqn. (40) holds for Algorithm 1 with any non-increasing step sizes. Assuming the path-length is known to be bounded as $\mathcal{P}_T(x_{1:T}^\star) \leq \mathcal{P}_T$, using the specific choice of the step size sequence $\eta_t = \frac{(D+1)(1+\mathcal{P}_t)}{\sqrt{2\sum_{\tau=1}^{t}(1+\mathcal{P}_\tau)||\nabla_\tau||^2}}$, $t \geq 1$, we can upper bound term (C) as follows

$$\frac{1}{2}\sum_{t=1}^{T} \eta_t ||\nabla_t||^2 = \frac{(D+1)}{2\sqrt{2}} \sum_{t=1}^{T} \frac{(1 + \mathcal{P}_t)||\nabla_t||^2}{\sqrt{2\sum_{\tau=1}^{t}(1+\mathcal{P}_\tau)||\nabla_\tau||^2}}$$

$$\leq \frac{(D+1)}{2\sqrt{2}} \int_0^{\sum_{t=1}^{T}(1+\mathcal{P}_t)||\nabla_t||^2} \frac{dz}{\sqrt{z}}$$

$$= \frac{(D+1)}{\sqrt{2}} \sqrt{\sum_{t=1}^{T}(1 + \mathcal{P}_t)||\nabla_t||^2}.$$

Hence, from (40), the dynamic regret for Algorithm 1 can be bounded as

$$\text{D-Regret}_T(\mathcal{P}_T) \leq \sqrt{2}(D+1)\sqrt{\sum_{t=1}^{T}(1 + \mathcal{P}_t)||\nabla_t||^2}. \tag{41}$$

# E. Deferred Technical Details Related to the Optimistic Setting

## E.1. Base Optimistic OCO Algorithm and Regret Guarantee

---

**Algorithm 5** Optimistic Online Mirror Descent

---

1: **Input** : Convex decision set $\mathcal{X}$, sequence of convex cost functions $\{\hat{f}_t\}_{t=1}^{T}$, $\text{diam}(\mathcal{X}) = D$.
2: **Initialize** : $x_1 \in \mathcal{X}$ arbitrarily.
3: **for** round $t = 1, 2, \ldots$ **do**
4:    Play action $x_t$, receive $\hat{f}_t$. Compute $\nabla_t = \nabla\hat{f}_t(x_t)$.
5:    Compute $\eta_{t+1} = \min\left\{\frac{\sqrt{\beta B}}{\sqrt{\mathcal{E}_t(\hat{f})} + \sqrt{\mathcal{E}_{t-1}(\hat{f})}}, \frac{\beta}{\max_{1 \leq \tau \leq t+1} \bar{L}_\tau}\right\}$
6:    $\tilde{x}_{t+1} := \arg\min_{x \in \mathcal{X}} \langle \nabla_t, x \rangle + \frac{1}{\eta_t} B^R(x; \tilde{x}_t)$.
7:    Compute $\bar{\nabla}_{t+1} = \nabla\bar{\hat{f}}_{t+1}(\tilde{x}_{t+1})$.
8:    $x_{t+1} := \arg\min_{x \in \mathcal{X}} \langle \bar{\nabla}_{t+1}, x \rangle + \frac{1}{\eta_{t+1}} B^R(x; \tilde{x}_{t+1})$.
9: **end for**

---

**Theorem 10** (Adapted from Theorem 10 of Lekeufack & Jordan (2025)). *Let Algorithm 5 be run on a sequence of cost functions $\{\hat{f}_t\}_{t=1}^T$. Let the corresponding sequence of predicted costs be $\{\tilde{\hat{f}}_t\}_{t=1}^T$. If $\bar{L}_t$ is the Lipschitz constant of the $t$-th predicted loss, then upon running Algorithm 5 with the following learning rate schedule*

$$\eta_t = \min\left\{\frac{\sqrt{\beta B}}{\sqrt{\mathcal{E}_{t-1}(\hat{f})} + \sqrt{\mathcal{E}_{t-2}(\hat{f})}}, \frac{\beta}{\max_{1 \leq \tau \leq t} \bar{L}_\tau}\right\} \text{ we get the following regret bound}$$

$$\mathsf{Regret}_t \leq 5\sqrt{\frac{B}{\beta}}\sqrt{\mathcal{E}_t(\hat{f})} + 5\frac{B}{\beta}\max_{1 \leq \tau \leq t}\bar{L}_\tau$$

*where $\beta$ is the strong convexity parameter of the regularizer $R$ defining the Bregman divergence, $B$ is an upper bound on this divergence over the algorithm's execution and $\mathcal{E}(\hat{f})$ has been defined in (23).*

**Remark 11.** *The original theorem statement from Lekeufack & Jordan (2025) assumed that $\bar{L}_t \leq \bar{L}_{t+1}$ and thus instead of $\max_{1 \leq \tau \leq t}\bar{L}_\tau$ it only had $\bar{L}_t$. The assumption naturally held true in their case since they made use of time-invariant Lyapunov functions as in Sinha & Vaze (2024) and this implied $\bar{L}_t = (1 + \Phi'(Q(t)))G$. In our setting, since $\bar{L}_t = (1 + \Phi'_t(Q(t)))G$, the Lipschitz constants may not be monotonically non-decreasing and to compensate for that, we replace the occurrence of $\bar{L}_t$ in the theorem statement with $\max_{1 \leq \tau \leq t}\bar{L}_\tau$.*

### E.2. Proof of Theorem 8

The proof follows a similar structure to the proof of Theorem 2 but incorporates the delayed queue update and the regret bound for the optimistic OCO algorithm.

Using similar analysis as before,

$$\Phi_{\tau+1}(Q(\tau+1)) - \Phi_\tau(Q(\tau)) \leq \Phi'_{\tau+1}(Q(\tau+1))\tilde{g}_\tau(x_\tau)$$

Let $x^* \in \mathcal{X}^*$ be a feasible comparator, so $g_\tau(x^*) \leq 0$ for all $\tau$. Consider the quantity $\Phi_{\tau+1}(Q(\tau+1)) - \Phi_\tau(Q(\tau)) + \tilde{f}_\tau(x_\tau) - \tilde{f}_\tau(x^*)$:

$$\begin{aligned}
&\Phi_{\tau+1}(Q(\tau+1)) - \Phi_\tau(Q(\tau)) + \tilde{f}_\tau(x_\tau) - \tilde{f}_\tau(x^*) \\
\leq&\Phi'_{\tau+1}(Q(\tau+1))\tilde{g}_\tau(x_\tau) + (x_\tau) + \tilde{f}_\tau(x_\tau) - \tilde{f}_\tau(x^*) \\
=&(\tilde{f}_\tau(x_\tau) + \Phi'_\tau(Q(\tau))\tilde{g}_\tau(x_\tau) - (\tilde{f}_\tau(x^*) + \Phi'_\tau(Q(\tau))\tilde{g}_\tau(x^*)) + \Phi'_{\tau+1}(Q(\tau+1))\tilde{g}_\tau(x_\tau) - \Phi'_\tau(Q(\tau))\tilde{g}_\tau(x_\tau) \\
=&\hat{f}_\tau(x_\tau) - \hat{f}_\tau(x^*) + \Phi'_{\tau+1}(Q(\tau+1))\tilde{g}_\tau(x_\tau) - \Phi'_\tau(Q(\tau))\tilde{g}_\tau(x_\tau)
\end{aligned}$$

Also note that,

$$\begin{aligned}
\Phi'_{\tau+1}(Q(\tau+1))\tilde{g}_\tau(x_\tau) - \Phi'_\tau(Q(\tau))\tilde{g}_\tau(x_\tau) &\leq \Phi'_{\tau+1}(Q(\tau+1)) - \Phi'_\tau(Q(\tau)) \\
&= \lambda_{\tau+1}(\Phi_{\tau+1}(Q(\tau+1)) + 1) - \lambda_\tau(\Phi_\tau(Q(\tau)) + 1) \\
&\leq \lambda_1(\Phi_{\tau+1}(Q(\tau+1)) - \Phi_\tau(Q(\tau))) \\
&\leq \frac{1}{4}(\Phi_{\tau+1}(Q(\tau+1)) - \Phi_\tau(Q(\tau)))
\end{aligned}$$

where the first inequality follows because $\tilde{g}_\tau(x_\tau) \leq 1$, the second equality follows due to the form of $\Phi_t$, the third inequality follows due to the decreasing nature of $\lambda_t$ and the last inequality follows because $\lambda_1 \leq \frac{1}{4}$. Therefore, we have,

$$\Phi_{\tau+1}(Q(\tau+1)) - \Phi_\tau(Q(\tau)) + \tilde{f}_\tau(x_\tau) - \tilde{f}_\tau(x^*) \leq \hat{f}_\tau(x_\tau) - \hat{f}_\tau(x^*) + \frac{1}{4}(\Phi_{\tau+1}(Q(\tau+1)) - \Phi_\tau(Q(\tau)))$$

Summing from $\tau = 1$ to $t$:

$$\Phi_{t+1}(Q(t+1)) - \Phi_1(Q(1)) + \sum_{\tau=1}^t(\tilde{f}_\tau(x_\tau) - \tilde{f}_\tau(x^*)) \leq \sum_{\tau=1}^t \hat{f}_\tau(x_\tau) - \hat{f}_\tau(x^*) + \frac{1}{4}\Phi_{t+1}(Q(t+1))$$

Since $Q(1) = 0$ and $\Phi_1(0) = 0$, and recognizing the regret terms $\text{Regret}_t(x^*) = \sum_{\tau=1}^{t}(\tilde{f}_\tau(x_\tau) - \tilde{f}_\tau(x^*))$ and $\text{Regret}'_t(x^*) = \sum_{\tau=1}^{t}(\hat{f}_\tau(x_\tau) - \hat{f}_\tau(x^*))$,

$$\frac{3}{4}\Phi_{t+1}(Q(t+1)) + \text{Regret}(x^\star) \le \text{Regret}'(x^\star) \tag{42}$$

where $\text{Regret}'_t(x^*)$ is the regret of the base optimistic OCO algorithm (Algorithm 5) run on the sequence of surrogate costs $\{\hat{f}_\tau\}_{\tau=1}^{t}$. We bound $\text{Regret}'_t(x^*)$ using Theorem 10:

$$\text{Regret}'_t(x^*) \le 5\sqrt{\frac{B}{\beta}}\sqrt{\mathcal{E}_t(\hat{f})} + 5\frac{B}{\beta}\max_{1\le\tau\le t}\bar{L}_\tau$$

Substituting this back:

$$\frac{3}{4}\Phi_{t+1}(Q(t+1)) + \text{Regret}_t(x^*) \le 5\sqrt{\frac{B}{\beta}}\sqrt{\mathcal{E}_t(\hat{f})} + 5\frac{B}{\beta}\max_{1\le\tau\le t}\bar{L}_\tau \tag{43}$$

First, we bound the surrogate prediction error $\mathcal{E}_t(\hat{f})$ using the definition of the prediction error:

$$\sqrt{\mathcal{E}_t(\hat{f})} = \sqrt{\sum_{\tau=1}^{t}(\hat{f}_\tau(x_\tau) - \bar{\hat{f}}_\tau(x_\tau))^2} = \sqrt{\sum_{\tau=1}^{t}(\tilde{f}_\tau + \Phi'_\tau(Q(\tau))\tilde{g}_\tau - \bar{f}_\tau - \Phi'_\tau(Q(\tau))\bar{g}_\tau)^2}$$

$$\le \sqrt{\sum_{\tau=1}^{t}2(\tilde{f}_\tau(x_\tau) - \bar{f}_\tau(x_\tau))^2 + \sum_{\tau=1}^{t}2\Phi'(Q(\tau))^2(\tilde{g}_\tau(x_\tau) - \bar{g}_\tau(x_\tau))^2} \le \sqrt{\sum_{\tau=1}^{t}2\epsilon_\tau(f) + \sum_{\tau=1}^{t}2\Phi'(Q(\tau))^2\epsilon_\tau(\tilde{g})}$$

$$\le \sqrt{2\mathcal{E}_t(f)} + (\Phi_t(Q(t)) + 1)\sqrt{\sum_{\tau=1}^{t}2\lambda_\tau^2\epsilon_\tau(\tilde{g})} \tag{44}$$

$$5\sqrt{\frac{B}{\beta}}\sqrt{\mathcal{E}_t(\hat{f})} \le 5\sqrt{\frac{B}{\beta}}\sqrt{2\mathcal{E}_t(f)} + (\Phi_t(Q(t)) + 1)5\sqrt{\frac{B}{\beta}}\sqrt{\sum_{\tau=1}^{t}2\lambda_\tau^2\epsilon_\tau(\tilde{g})}$$

We choose $\lambda_\tau$ adaptively based on the prediction error of the constraints as defined before Theorem 8. Recall that,

$$\lambda_\tau = \frac{1}{20(\sqrt{\frac{B}{\beta}} + \frac{B}{\beta})\sqrt{\gamma_\tau + 1}\sqrt{\log(\gamma_\tau + 1) + 1}(\log(\log(\gamma_\tau + 1) + 1) + 1)}$$

. Note that, $\epsilon_\tau(\tilde{g}) \le D^{-2}$ (as pre-processed functions are $(2D)^{-1}$-Lipschitz), we can show that $\mathcal{E}_\tau(\tilde{g}) = \mathcal{E}_{\tau-1}(\tilde{g}) + \epsilon_\tau(\tilde{g}) \le \mathcal{E}_{\tau-1}(\tilde{g}) + D^{-2} \le \gamma_\tau$. Thus:

$$\lambda_\tau \le \frac{1}{20(\sqrt{\frac{B}{\beta}} + \frac{B}{\beta})\sqrt{\mathcal{E}_\tau(\tilde{g}) + 1}\sqrt{\log(\mathcal{E}_\tau(\tilde{g}) + 1) + 1}(\log(\log(\mathcal{E}_\tau(\tilde{g}) + 1) + 1) + 1)}$$

$$\implies \lambda_\tau^2 \le \frac{1}{400(\sqrt{\frac{B}{\beta}} + \frac{B}{\beta})^2(\mathcal{E}_\tau(\tilde{g}) + 1)(\log(\mathcal{E}_\tau(\tilde{g}) + 1) + 1)(\log(\log(\mathcal{E}_\tau(\tilde{g}) + 1) + 1) + 1)^2}$$

$$\implies 2\epsilon_\tau(\tilde{g})\lambda_\tau^2 \le \frac{\epsilon_\tau(\tilde{g})}{200(\sqrt{\frac{B}{\beta}} + \frac{B}{\beta})^2(\mathcal{E}_\tau(\tilde{g}) + 1)(\log(\mathcal{E}_\tau(\tilde{g}) + 1) + 1)(\log(\log(\mathcal{E}_\tau(\tilde{g}) + 1) + 1) + 1)^2}$$

$$\implies 2\epsilon_\tau(\tilde{g})\lambda_\tau^2 \le \frac{(\mathcal{E}_\tau(\tilde{g}) + 1) - (\mathcal{E}_{\tau-1}(\tilde{g}) + 1)}{200(\sqrt{\frac{B}{\beta}} + \frac{B}{\beta})^2(\mathcal{E}_\tau(\tilde{g}) + 1)(\log(\mathcal{E}_\tau(\tilde{g}) + 1) + 1)(\log(\log(\mathcal{E}_\tau(\tilde{g}) + 1) + 1) + 1)^2}$$

. The above form makes it clear that $\sum_{\tau=1}^{t} 2\lambda_\tau^2 \epsilon_\tau(\tilde{g})$ is amenable to the integral-sum trick as in Lemma 9. Overall, we get,

$$\sum_{\tau=1}^{t} 2\lambda_\tau^2 \epsilon_\tau(\tilde{g}) \leq \frac{1}{100(\sqrt{\frac{B}{\beta}} + \frac{B}{\beta})^2} \implies 5\sqrt{\frac{B}{\beta}}\sqrt{\sum_{\tau=1}^{t} 2\lambda_\tau^2 \epsilon_\tau(\tilde{g})} \leq \frac{1}{2}$$

Overall, we get that,

$$\sqrt{\mathcal{E}_t(\hat{f})} \leq 5\sqrt{\frac{B}{\beta}}\sqrt{2\mathcal{E}_t(f)} + \frac{1}{2}\Phi_t(Q(t)) + \frac{1}{2}$$

Then, we bound $\max_{1 \leq \tau \leq t} \bar{L}_\tau$,

$$\max_{1 \leq \tau \leq t} \bar{L}_t \leq (2D)^{-1}\max_{1 \leq \tau \leq t}(1 + \lambda_\tau e^{\lambda_\tau Q(\tau)}) \leq (2D)^{-1} + (2D)^{-1}\max_{1 \leq \tau \leq t}\lambda_\tau e^{\lambda_\tau Q(\tau)} \leq (2D)^{-1} + (2D)^{-1}\lambda_1 e^{\lambda_t Q(t)}$$

Note that, $5\frac{B}{\beta}\lambda_1 \leq \frac{1}{4D^{-1}}$, so $(2D)^{-1}5\frac{B}{\beta}\lambda_1 \leq \frac{1}{8}$. Overall, we get,

$$5\frac{B}{\beta}\max_{1 \leq \tau \leq t} \bar{L}_t \leq 5\frac{B}{\beta}(2D)^{-1} + \frac{1}{8}(\Phi_t(Q(t)) + 1)$$

Substituting the upper-bound of $5\sqrt{\frac{B}{\beta}}\sqrt{\mathcal{E}_t(\hat{f})}$ and $5\frac{B}{\beta}\max_{1 \leq \tau \leq t} \bar{L}_\tau$ in inequality (43):

$$\frac{3}{4}\Phi_{t+1}(Q(t+1)) + \text{Regret}_t(x^*) \leq 5\sqrt{\frac{B}{\beta}}\sqrt{2\mathcal{E}_t(f)} + \frac{1}{2}\Phi_t(Q(t)) + \frac{1}{2} + 5\frac{B}{\beta}(2D)^{-1} + \frac{1}{8}(\Phi_t(Q(t)) + 1)$$

Further, note that, $\Phi_t(Q(t)) \leq \Phi_{t+1}(Q(t+1))$,

$$\frac{3}{4}\Phi_{t+1}(Q(t+1)) + \text{Regret}_t(x^*) \leq 5\sqrt{\frac{B}{\beta}}\sqrt{2\mathcal{E}_t(f)} + \frac{5}{8} + 5\frac{B}{\beta}(2D)^{-1} + \frac{5}{8}\Phi_{t+1}(Q(t+1))$$

$$\frac{1}{8}\Phi_{t+1}(Q(t+1)) + \text{Regret}_t(x^*) \leq 5\sqrt{\frac{B}{\beta}}\sqrt{2\mathcal{E}_t(f)} + \frac{5}{8} + 5\frac{B}{\beta}(2D)^{-1}$$

Dropping the non-negative $\Phi_t(Q(t))$ term gives the regret bound for the pre-processed functions:

$$\text{Regret}_t(x^*) \leq 5\sqrt{\frac{B}{\beta}}\sqrt{2\mathcal{E}_t(f)} + \frac{5}{8} + 5\frac{B}{\beta}(2D)^{-1}.$$

For the CCV bound, use the trivial lower bound $\text{Regret}_t(x^*) \geq -Dt/(2D) \cdot \alpha = -t/2$. Substitute this into the inequality before dropping the $\Phi$ term:

$$\frac{1}{8}\Phi_{t+1}(Q(t+1)) \leq 5\sqrt{\frac{B}{\beta}}\sqrt{2\mathcal{E}_t(f)} + \frac{5}{8} + 5\frac{B}{\beta}(2D)^{-1} + \frac{t}{2}.$$

$$\Phi_{t+1}(Q(t+1)) \leq 40\sqrt{\frac{B}{\beta}}\sqrt{2\mathcal{E}_t(f)} + 10 + 20\frac{B}{\beta D} + 4t.$$

Since $\Phi_t(x) = e^{\lambda_t x} - 1$:

$$e^{\lambda_{t+1}Q(t+1)} \leq 40\sqrt{\frac{B}{\beta}}\sqrt{2\mathcal{E}_t(f)} + 11 + 20\frac{B}{\beta D} + 4t.$$

$$Q(t+1) \leq \frac{1}{\lambda_{t+1}} \log(40\sqrt{\frac{B}{\beta}} \sqrt{2\mathcal{E}_t(f)} + 11 + 20\frac{B}{\beta D} + 4t)$$

$$Q(t+1) \leq 20\left(\sqrt{\frac{B}{\beta}} + \frac{B}{\beta}\right)\sqrt{\gamma_{t+1}+1}\sqrt{\log(\gamma_{t+1}+1)+1}$$
$$\times \left(\log(\log(\gamma_{t+1}+1)+1)+1\right)$$
$$\times \log\left(40\sqrt{\frac{B}{\beta}} \sqrt{2\mathcal{E}_t(f)} + 11 + 20\frac{B}{\beta D} + 4t\right).$$

where $\gamma_{t+1} = \mathcal{E}_t(\tilde{g}) + D^{-2}$.

Overall, we get a bound of $O(\sqrt{\mathcal{E}_t(\tilde{f})})$ for regret and a bound of $\tilde{O}(\sqrt{\mathcal{E}_t(\tilde{g})})$ for the CCV.

## F. Numerical Simulations

**Setup:** We consider a constrained version of the online shortest path problem (Hazan, 2022), where the length corresponds to the latency and the constraint is on the long-term cumulative bandwidth across the path. In this problem, on each round, the online algorithm first selects a route connecting a source $s$ to a destination $d$ on a graph $G(V, E)$. The latency and bandwidth of each edge vary across rounds, reflecting dynamic network conditions. The objective is to minimize the cumulative latency subject to a long-term lower bound on the cumulative bandwidth. Formally, the following sequence of events takes place on the $t^{\text{th}}$ round:

1. The algorithm first chooses a route (randomly or otherwise) $p_t \in P_{s,d}$, where $P_{s,d}$ is the set of all $s - d$ routes in the graph.

2. A latency of $\tau_e(t)$ and a bandwidth of $l_e(t)$ is chosen by an adversary for each edge $e \in E$.

3. The algorithm incurs a latency cost of $\sum_{e \in p_t} \tau_e(t)$ and a bandwidth cost of $-\sum_{e \in p_t} l_e(t)$ on round $t$.

We represent each route $p$ by its corresponding $|E|$-dimensional binary incidence vector where $p_e = 1$ if the edge $e \in E$ belongs to the route or $p_e = 0$ otherwise. On each round, our online policy returns an element from the convex hull of $P_{s,d}$, also known as the unit flow polytope (Hazan, 2022). We use Dijkstra's algorithm for computing the weighted shortest route. It is well-known that any element in the unit flow polytope can be efficiently decomposed into a convex combination of at most $|E|$ number of $s - d$ routes using the flow decomposition lemma (Williamson, 2019, Lemma 2.20). These convex combinations can be used to randomly select a single route on each round, incurring the same expected cost. The experiments are performed on a quad-core CPU with 8 GB RAM. The experimental setup mostly follows that in Sarkar et al. (2025).

**Dataset:** For the experiments, we utilize a semi-synthetic dataset designed to simulate dynamic network conditions, constructed following methodologies established in prior work (Sarkar et al., 2025) for evaluating online network algorithms. The dataset originates from raw data collected via public probes on the RIPE Atlas global network measurement platform (Staff, 2015), specifically HTTPS measurements capturing bandwidth and latency between network nodes. The resulting graph consists of $n = 191$ nodes and $m = 1200$ edges, where each edge is characterized by latency and bandwidth attributes. The construction involves centering the graph around a hub node, adding connections based on the RIPE Atlas data, and then introducing additional random edges, with edge weights determined by latencies relative to the hub and minimum bandwidths between connected nodes. To mimic real-world fluctuations for the online setting, temporal variations are introduced through random scaling factors applied at each iteration: latency values are scaled randomly between 0.5x and 1.5x, and bandwidth values between 0.8x and 1.2x. This process generates the time-varying latency and bandwidth matrices used for evaluating algorithm performance over a horizon length of $T = 1600$ iterations. This dataset provides a realistic and challenging testbed for constrained online optimization algorithms.

The two-panel plot in Figure 1 reports both cumulative constraint violation and cumulative regret.

**Results:**   The empirical results presented in Figure 1 provide strong validation for our theoretical contributions. The plots compare the performance of our proposed anytime algorithm against two key baselines: the standard fixed-horizon algorithm from Sinha and Vaze (2024), which assumes prior knowledge of the total horizon $T$, and its adaptation to an unknown horizon setting via the standard doubling trick.

In summary, the simulations reveal a key practical advantage of our approach, demonstrating that our anytime algorithm outperforms both the fixed-horizon and doubling trick baselines.

The doubling-trick-based method performs the worst. This is due to its restarting technique, where the algorithm discards the information it has accumulated up to that point and starts from scratch. Our observations match those in (Besson & Kaufmann, 2018) where they evaluated several versions of the doubling trick and concluded that performance is substantially worse than that of an intrinsically anytime algorithm, even one with less favorable theoretical guarantees. Moreover, this performance gap becomes larger each time the algorithm is restarted.

While the fixed-horizon method is optimal in a worst-case theoretical sense, its use of a single, conservative trade-off parameter hinders its practical performance. Our algorithm, with its time-varying parameter, is more adaptive; it aggressively penalizes violations in the early stages, quickly learning the constraints and achieving a superior performance trajectory. This confirms that our method is not just a tool for handling an unknown horizon, but a fundamentally more effective strategy that also avoids the instability and performance loss inherent in the doubling trick.

