# OpenReview forum: "Optimal Anytime Algorithms for Online Convex Optimization with Adversarial Constraints"
_ICML.cc/2026/Conference — ICML 2026 regular_

### Official Review · Reviewer_2Efb · 2026-02-17

**Soundness:** 4
**Presentation:** 4
**Significance:** 3
**Originality:** 3
**Overall Recommendation:** 5
**Confidence:** 3

**Summary:**

This paper presents an anytime online algorithm that considers an adversarial cost and constraint functions. The unknown time horizon case is considered. Compared to prior works, the paper retains $O(\sqrt{t})$ regret and also improves the CCV violation from $O(\sqrt{T})$ to $O(\sqrt{t})$ for each $t$.

**Compliance With Llm Reviewing Policy:**

Affirmed.

**Final Justification:**

The paper presents a clear theoretical analysis with good contribution, and the rebuttal adequately clarifies my concerns. I recommend an acceptance of this paper.

**Key Questions For Authors:**

I hope that Questions 8 and 11 are well-addressed. All others are minor comments.

1. In section 4.3.2, the authors should perhaps use just Regret_t rather than Regret_t (x*). Based on Eqn (2), x* is already well-defined so you don’t need to incorporate into the argument.

2. In Line 257 (right column), after ‘Hence, Eqn.(16) implies that’, you should have $\exp(\lambda_t Q(t)) \le t+2\sqrt{t} + 2$, instead of $1$, since you have $ \exp(\lambda_t Q(t))  =  \Phi_t(Q(t)) + 1$, so all the analysis may change from $4t$ to $5t$ (although it does not affect the order).

3. In Remark 3, the authors said that they don’t require information on G, but instead they need information about $\nabla f_t$ and $\nabla g_t$’s. This seems to be more restrictive since it requires the information on gradients, so I recommend that the authors remove the ‘would not need knowledge of G’ part. Still, I appreciate the contents on Appendix C, and it is valuable itself.

4. Using $\hat f_t$ again in Section 5.1 is not recommended, as it may confuse readers with the notations in the proof of Section 4.

5. 19 in Line 328 should be (19).

6. $D-Regret_T (P_T)$ in Theorem 4, 6, or the expression in (17), or the notations in Section 5.2 do not match.

7. I greatly recommend putting some experiments on the main content, instead of putting everything in the appendix.

8. I appreciate all the other analysis, but I’m not quite sure whether Theorem 4 is novel. It must be a very simple tweak to improve $T P_T$ to $\sum_{t=1}^T P_t$.. What’s so special about this analysis? Is it special because the authors consider a constrained optimization problem?

9. $\\{x_t^\star\\}$ in Line 377 should be $\\{x_\tau^\star\\}$.

10. In the bottom of page 7, Appendix D.2 is not hyperlinked.

11. Intuitively, the proposed regret and CCV bounds seem to be optimal. Can you add a remark on whether the regret and CCV violation $O(\sqrt{t})$ are optimal (especially on CCV since this is the main improvement of this paper)? This would greatly improve the significance of the work.

**Limitations:**

Yes.

**Strengths And Weaknesses:**

Strengths: The construction of the sequence $\lambda_t$ is simple and novel, and the analysis in the main content is all correct.  In addition, the paper presents dynamics regret analysis and optimistic setting as well.

Weaknesses: Since the whole analysis is on the simple construction of $\lambda_t$, the experiments should be included in the main content to persuade the readers.

---

> ### Author Rebuttal · Authors · 2026-03-27
>
> We thank the reviewer for a detailed review of our paper.
>
> **On the suggestions:**
>
>  In the revised version of the paper, we will incorporate the excellent suggestions (points 1, 4 and 7) and correct the minor errors in presentation (points 2, 5, 6, 9 and 10).
>
> **Q3. On the knowledge of observed gradients vs knowledge of $G$:**
>
> We believe there may be a slight confusion regarding the notation. Here, $G$ denotes a uniform upper bound on the Lipschitz constants of \emph{all} cost and constraint functions, which are chosen by the adversary. While this is a standard assumption in the online learning literature, it effectively constitutes counterfactual knowledge; in practice, a tight upper bound on $G$ may be unavailable. This motivates the design of adaptive algorithms that do not rely on such prior information. In Appendix C, we show that our proposed framework admits an extension to precisely this setting.
>
> In the first-order OCO framework we consider, receiving local gradients $\nabla f_t(x_t)$ and $\nabla g_t(x_t)$ after playing action $x_t$ is entirely standard [1, 2]. Indeed, this is analogous to classical first-order methods, such as gradient descent, which operate using local gradient information.
>
> In summary, requiring a priori knowledge of a global upper bound $G$ on the Lipschitz constants across the entire horizon is a significantly stronger, and often impractical assumption. In contrast, dynamically adapting to observed local gradients, without ever needing to know $G$ in advance, constitutes a strict relaxation of this requirement. To avoid any potential confusion, we will add a brief clarifying sentence to Remark 3 emphasizing that gradient observation is standard first-order feedback. We also note that the $G$-dependent algorithm of Sinha \& Vaze (2024) relies on both $G$ and gradient information.
>
> **W1 \& Q7: Moving Experiments to the Main Text**
>
> We agree with this excellent suggestion. In the camera-ready version, we will use the additional space to move the numerical simulations (Figures 1 and 2) from Appendix F into the main text (Section 7), thereby better highlighting the practical advantages of our algorithm over the doubling trick.
>
> **Q8: Novelty of Theorem 4:**
>
> Note that Theorem 4 pertains to the unconstrained OCO setting and serves as a crucial bridge in our constrained-to-unconstrained reduction. Its novelty in our context lies in explicitly incorporating the time-varying path length $\mathcal{P_t}$ into the Lyapunov-based surrogate loss framework (see Eq. (21)). The key point here is that while the terminal path length $P_T$ is only available offline at the end of the horizon, the instantaneous quantities $\(\mathcal{P_t}\)_{t \geq 1}$ can be computed online and can be used for tuning the step sizes in an adaptive fashion. This careful integration enables us to obtain anytime dynamic regret guarantees in the presence of adversarial constraints, without requiring prior knowledge of $P_T$.
>
> On the technical side, the standard static regret analysis of AdaGrad (e.g., Orabona, 2019, Theorem 4.14) does not directly apply to Theorem 4. The key difficulty is that term (B) in Eq. (38) cannot be bounded in a straightforward manner due to the time-varying nature of the benchmark sequence ${x_t^*}$. Consequently, establishing the regret bound requires several non-trivial steps that crucially exploit the monotonicity of the step sizes.
>
> We will clarify these technical points more explicitly in the revised version.
>
> **Q11: Optimality of the $\mathcal{O}(\sqrt{t})$ bounds:**
>
> The bounds presented in this paper are minimax optimal. In particular, Theorem 3 of [3] establishes that no online algorithm can achieve better than $\tilde{O}(\sqrt{T})$ regret and CCV for the COCO problem, even in the fixed-horizon setting of length $T$. Since our guarantees hold simultaneously for all $t \geq 1$, the resulting anytime bounds are order-optimal, i.e., tight up to logarithmic factors in $t$.
>
> We will include an explicit remark highlighting this point in the revised version. We also refer the reader to our response to Reviewer nWdw for further discussion.
>
> **References**
>
> [1] Orabona, Francesco. “A modern introduction to online learning.” arXiv preprint arXiv:1912.13213 (2019).
>
> [2] Hazan, Elad. "Introduction to online convex optimization." Foundations and Trends in Optimization 2, no. 3-4 (2016): 157-325.
>
> [3] Sinha, Abhishek, and Rahul Vaze. "Optimal algorithms for online convex optimization with adversarial constraints." Advances in Neural Information Processing Systems 37 (2024): 41274-41302.

---

> > ### Author Rebuttal · Reviewer_2Efb · 2026-03-31
> >
> > Thanks for addressing my questions, and I look forward to the updated draft based on my comments. I will keep my positive score.

---

### Official Review · Reviewer_roHe · 2026-02-17

**Soundness:** 2
**Presentation:** 3
**Significance:** 3
**Originality:** 3
**Overall Recommendation:** 4
**Confidence:** 4

**Summary:**

This paper proposed practical anytime online convex optimization algorithms with constraints without doubling trick / restarts. This design belongs to the general approach in this field by designing lyapunov potential function and a surrogate loss that simultaneous capture regret incurred by the cost functions and cumulative values due to violation of constraints. Previous work does not achieve anytime regret and this paper proposed a remedy to achieve anytime guarantee by a time-varying lyapunov potential function with careful reweighing to preserve technical monotonicity for deriving regret. Overall, the sublinear regret and cumulative violation value were achieved up to log factors. And the notion of regret can also be extended to optimistic, dynamic settings.

**Compliance With Llm Reviewing Policy:**

Affirmed.

**Final Justification:**

As my primary concern was requiring both $G,D$ in the main text, the author has pointed out that appendix C is a variants that eliminate the dependence on $G$. So the problem practically only have dependence on $D$, which is satisfactory in OCO.

**Key Questions For Authors:**

I wonder whether there is lower bound or any type of justices on the problem dependent parameters requirements. For example, without restart, knowing $G$ and $D$ at the same time is necessary.

**Limitations:**

Yes

**Strengths And Weaknesses:**

Strength:
- the paper is well written, motivation and the necessity of an anytime guarantee without restarting is well supported.
- the intuition of the design of the time varing lyapunov potential, necessity of reweighing as a correction and surrogate loss construction was well explained.
- the paper indeed achieved its claim: anytime guarantee without restarts with matching rates to previous works.
- the metric contains two parts, one of them is regret on cost functions. this paper extends also to dynamic regret and optimistic regret while considering the constraints violation

Weakness:
- in general the reliance on problem-dependent parameters is restrictive. Although the authors note that prior work has similar dependencies. For example, the requirement of any path length $P_t$ rather than total path length $P_T$, which might seem as a gain. But this is fundamentally a consequence that the framework is compatible to time varying learning rate, which is due to the time varying lyapunov potential. This requires both  $G$ and $D$ at the same time. As a result, the claimed gain comes at the expense of requiring stronger knowledge of problem-dependent quantities.

---

> ### Author Rebuttal · Authors · 2026-03-27
>
> We thank the reviewer for the review. We will address the parameter dependence on G and D.
>
> **1. Removing the dependence on $G$:**
>
> Please note that the paper addresses precisely this question in Appendix C, titled *“Lipschitz Adaptive COCO Algorithm”* (see Remark 3 in the main text). In brief, by making both the step size and the time-varying Lyapunov parameters adaptive and dependent on the observed gradient norms, our framework achieves data-dependent bounds of $\mathcal{O}\left(\sqrt{\sum_{\tau=1}^t ||\nabla f_\tau(x_\tau)||2^2}\right)$ for regret and $\tilde{\mathcal{O}}\left(\sqrt{\sum_{\tau=1}^t ||\nabla g_\tau^+(x_\tau)||_2^2}\right)$ for CCV.
>
> This adaptive extension completely removes the need to know $G$ (or an upper bound on it) in advance, while crucially preserving the anytime property and avoiding the doubling trick. We will emphasize this important generalization more clearly in the revised version.
>
> **2. The theoretical necessity of $D$:**
>
> Compared to [1], the present work removes the need to know the horizon $T$ and the Lipschitz constant $G$, while still achieving order-optimal guarantees. However, our algorithm continues to rely on prior knowledge of the domain diameter $D$.
>
> There has been substantial recent progress on *parameter-free* methods that aim to eliminate such dependencies [2,3]. In principle, one might hope to leverage parameter-free variants of AdaGrad to remove the need for $D$. A key technical obstacle, however, is that our surrogate cost function $\hat{f_t}$ depends explicitly on $D$ — see the definition of the parameter $\lambda_\tau$ following Eq. (32). As a result, it is not immediately clear how to adapt either the algorithm or the analysis to eliminate this dependence.
>
> In the revised version, we will include a dedicated “Limitations and Open Problems” section to highlight this issue and frame fully parameter-free COCO as an important direction for future work. We also refer the reader to our response to Reviewer nWdw for additional discussion.
>
> **References**
>
> [1] Sinha, Abhishek, and Rahul Vaze. "Optimal algorithms for online convex optimization with adversarial constraints." Advances in Neural Information Processing Systems 37 (2024): 41274-41302.
>
> [2] Orabona, Francesco. “A modern introduction to online learning.” arXiv preprint arXiv:1912.13213 (2019).
>
> [3] Jacobsen, Andrew, and Ashok Cutkosky. “Parameter-free mirror descent.” In Conference on Learning Theory, pp. 4160-4211. PMLR, 2022.

---

> > ### Author Rebuttal · Reviewer_roHe · 2026-04-03
> >
> > thank you for addressing my concern, i have updated the score.

---

### Official Review · Reviewer_5aHm · 2026-03-10

**Soundness:** 3
**Presentation:** 3
**Significance:** 3
**Originality:** 3
**Overall Recommendation:** 4
**Confidence:** 4

**Summary:**

This paper studies constrained online convex optimization with adversarial constraints in the anytime setting. The main technical contribution is an anytime algorithm that achieves $\tilde O(\sqrt{t})$ regret guarantees both for cumulative constraint violation and regret without requiring prior knowledge of the horizon. From my reading, the key novelty is the construction of time-varying Lyapunov functions together with the associated virtual queue $Q_t$ quantity, which allows the analysis to go through in the anytime setting. I think this is the most interesting part of the paper, and it provides a nontrivial theoretical insight beyond a routine adaptation of existing primal-dual methods.

**Compliance With Llm Reviewing Policy:**

Affirmed.

**Key Questions For Authors:**

1. Is the proposed framework able to explicitly trade off regret and constraint violation? For example, is it possible to tune the method toward achieving smaller regret (possibly even negative regret performance) at the cost of allowing larger constraint violation, or vice versa?

**Limitations:**

Yes

**Strengths And Weaknesses:**

Strengths:

1. The paper considers online convex optimization with adversarial constraints, and focuses on designing an algorithm that is anytime, i.e., does not need the horizon $T$ in advance.

2. The introduction of a sequence of time-varying Lyapunov functions, instead of a fixed one depending on the horizon, is quite neat. Related to this, the modified queue quantity $Q_t$ is novel and technically meaningful, which restores a monotonicity structure for the proof.  Besides, the proofs are also clean.

3. The paper also discusses extensions to dynamic regret and optimistic variant which avoids doubling tricks.


Weaknesses:

1. The discussion of related work is incomplete. For example, there are some relevant but missing works:

[1] Beyond Primal-Dual Methods in Bandits with Stochastic and Adversarial Constraints

[2] No-Regret is not enough! Bandits with General Constraints through Adaptive Regret Minimization

2. The parameter design requires the knowledge of $G$. I am not fully sure how strong or restrictive this assumption is.

3. I may have spotted a possible typo or inconsistency: in Eq. (17), it seems it should be $f_t$ instead of $\hat{f}_t$

---

> ### Author Rebuttal · Authors · 2026-03-27
>
> W1: Missing Related Work:
>
> Thank you for pointing these out. While both of these excellent works tackle adversarial constraints, they focus specifically on the partial-information multi-armed bandit setting. In contrast, our work addresses the full-information Online Convex Optimization (OCO) problem over continuous domains. From the methodological side, [2] relies on a primal-dual framework by enforcing weak adaptivity on both the primal and dual regret minimizers, whereas [1] abandons primal-dual methods entirely in favor of optimistic constraint estimations. Similarly to [1], our approach bypasses explicit dual-variable updates; however, instead of relying on optimistic bounds, we directly embed the constraints into a sequence of primal surrogate loss functions via time-varying Lyapunov functions to achieve our anytime guarantees. In the revised version, we will discuss both [1] and [2] in our related works section to provide a more comprehensive literature overview.
>
> W2: Parameter design requires knowledge of $G$:
>
> Please note that the paper addresses this question in Appendix C, titled *“Lipschitz Adaptive COCO Algorithm”* (see Remark 3 in the main text). In brief, by making both the step size and the time-varying Lyapunov parameters adaptive and dependent on the observed gradient norms, our framework achieves data-dependent bounds of $\mathcal{O}\left(\sqrt{\sum_{\tau=1}^t ||\nabla f_\tau(x_\tau)||2^2}\right)$ for regret and $\tilde{\mathcal{O}}\left(\sqrt{\sum_{\tau=1}^t ||\nabla g_\tau^+(x_\tau)||_2^2}\right)$ for CCV.
>
> This adaptive extension completely removes the need to know $G$ (or an upper bound on it) in advance, while crucially preserving the anytime property and avoiding the doubling trick. We will emphasize this important generalization more clearly in the revised version.
>
> W3: Typo in Eq. (17):
>
> Thank you for catching this typographical error. We will correct this typo in the revised manuscript.
>
> Q1: Trade-off between Regret and CCV:
>
> Even in the standard unconstrained OCO setting, Regret is information-theoretically lower bounded by $\Omega(\sqrt{t})$. Therefore, it is not possible to reduce regret below $O(\sqrt{t})$ by trading off with increased CCV. On the other hand, it is natural to consider reducing CCV at the expense of regret; this trade-off has indeed been explored in [3]. Extending such approaches to obtain anytime guarantees remains an interesting direction for future research.
>
> **References**
>
> [1] Bernasconi, Martino, Matteo Castiglioni, Andrea Celli, and Federico Fusco. "Beyond primal-dual methods in bandits with stochastic and adversarial constraints." Advances in Neural Information Processing Systems 37 (2024): 8541-8568.
>
> [2] Bernasconi, Martino, Matteo Castiglioni, and Andrea Celli. "No-Regret is not enough! Bandits with General Constraints through Adaptive Regret Minimization." In International Conference on Machine Learning, pp. 3877-3898. PMLR, 2025.
>
> [3] Sinha, Abhishek, and Rahul Vaze. "Beyond $\tilde {O}(\sqrt {T}) $ Constraint Violation for Online Convex Optimization with Adversarial Constraints." In The Thirty-ninth Annual Conference on Neural Information Processing Systems.

---

> > ### Author Rebuttal · Reviewer_5aHm · 2026-04-01
> >
> > Thank the authors for the detailed response. I will keep my score.

---

### Official Review · Reviewer_nWdw · 2026-03-12

**Soundness:** 3
**Presentation:** 4
**Significance:** 3
**Originality:** 3
**Overall Recommendation:** 5
**Confidence:** 4

**Summary:**

The paper gives an anytime extension of the regret guarantees shown by Sinha & Vaze '24 for COCO (OCO with adversarial constraints). The results are not obtained via a standard doubling trick (as shown in the appendix, this does not give the desired bounds), but via a reduction to AdaGrad. This reduction relies on a new cost function $\hat f_t$, a linear combination of the loss function $f_t$ and constraint function $(g_t)^+$. Regret guarantees are shown via a relating the AdaGrad regret guarantees for $\hat f_t$ to the original functions (regrets) and the cumulative constraint violations (CCV).
The results are extended to dynamic regret (for the loss function) and optimistic algorithms (obtaining the improvment from gradient norm/Lipschitz parameter to the prediction error).

**Compliance With Llm Reviewing Policy:**

Affirmed.

**Key Questions For Authors:**

Q1: Can the constant $D$ be improved to depend on $D*$ := diameter$(X*)$? (e.g. what if $D \propto T$ but $D* = 1$)
Q2: Are the results of Theorem 2 for CCV tight (up to polylog factors)?

**Limitations:**

This is theoretical work, thus no negative societal impract expected.

**Strengths And Weaknesses:**

Strengths:
- This is a well written paper. I realy appriciate that the authors do focus on communicating the main ideas in the cleanest and simplest case.
- The algorithmic and analysis ideas are nice, clean and non-trivial.

Weakness:
- No limitations are discussed.

---

> ### Author Rebuttal · Authors · 2026-03-27
>
> Q1: Can the constant $D$ be improved to depend on $D^\star := \text{diameter}(\mathcal{X}^\star)$?
>
> We thank the reviewer for this insightful question. In the standard OCO setting, regret depends on $D$ because the algorithm projects onto the known feasible set $\mathcal{X}$. In our COCO setting, the ''true" feasible set $\mathcal{X}^\star$ (set of all actions satisfying all $T$ constraints) is defined by adversarial constraints revealed online. Because the algorithm cannot project onto $\mathcal{X}^*$ at step $t$ (since future constraints are unknown), the base AdaGrad subroutine must project onto the global set $\mathcal{X}$. Consequently, the regret naturally scales with $D = \text{diam}(\mathcal{X})$. Achieving bounds scaling purely with $D^\star$ without prior knowledge of the constraints is an interesting open problem. One possible approach to obtaining such a bound is to employ parameter-free variants of AdaGrad [1,2], which do not require prior knowledge of the domain diameter. We leave a detailed investigation of this direction to future work. See also our response to the reviewer roHe.
>
>
>
> Q2: Are the results of Theorem 2 for CCV tight (up to polylog factors)?
>
> Yes, the bounds are minimax optimal up to polylogarithmic factors. A joint $\Omega(\sqrt{T})$ lower bound for both regret and cumulative constraint violation (CCV) was established in [3, Theorem 3] for the setting where the horizon $T$ is known in advance. Since our results achieve this rate simultaneously for all $t$, the resulting anytime CCV bound is order-optimal up to polylogarithmic factors. We will include this clarification as a remark in the revised version.
>
> W1: Limitations Section.
>
> You rightly pointed out the absence of a limitations section. In the revised version, we will include a dedicated “Limitations and Open Problems” paragraph, where we will explicitly discuss the algorithm’s reliance on knowing the domain diameter $D$.
>
> **References**
>
> [1] Orabona, Francesco. “A modern introduction to online learning.” arXiv preprint arXiv:1912.13213 (2019).
>
> [2] Jacobsen, Andrew, and Ashok Cutkosky. “Parameter-free mirror descent.” In Conference on Learning Theory, pp. 4160-4211. PMLR, 2022.
>
> [3] Sinha, Abhishek, and Rahul Vaze. "Optimal algorithms for online convex optimization with adversarial constraints." Advances in Neural Information Processing Systems 37 (2024): 41274-41302.

---

> > ### Author Rebuttal · Reviewer_nWdw · 2026-04-02
> >
> > I am fully satisfied with the author's answers and will keep my score recommending acceptance of the work.

---

### Decision · Program_Chairs · 2026-04-30

**Decision:**

Accept (regular)

**Comment:**

This paper gives an anytime algorithm for online convex optimization with adversarial constraints, where the horizon is unknown. The main technical contribution is a time-varying, horizon-oblivious Lyapunov function together with a modified queue recursion, which allows the authors to obtain regret and cumulative constraint violation guarantees that scale with the current time $t$, without using doubling tricks or restarts. The framework is also extended to dynamic-regret and optimistic settings.

The reviews were overall positive and viewed the core idea and analysis as novel and technically solid. The main concerns were mostly about presentation, related work, and parameter dependence, and these were satisfactorily addressed in the rebuttal, including clarification of the adaptive variant and the optimality discussion.